# The 3D Navier–Stokes Equations: Invariants, Local and Global Solutions

## Vladimir I. Semenov

Institute Physics, Mathematics and Computer Sciences, Baltic Federal University, Kaliningrad 236016, Russia; visemenov@rambler.ru

**Abstract:** In this article, I consider local solutions of the 3D Navier–Stokes equations and its properties such as an existence of global and smooth solution, uniform boundedness. The basic role is assigned to a special invariant class of solenoidal vector fields and three parameters that are invariant with respect to the scaling procedure. Since in spaces of even dimensions the scaling procedure is a conformal mapping on the Heisenberg group, then an application of invariant parameters can be considered as the application of conformal invariants. It gives the possibility to prove the sufficient and necessary conditions for existence of a global regular solution. This is the main result and one among some new statements. With some compliments, the rest improves well-known classical results.

**Keywords:** Navier–Stokes equations; global solutions; regular solutions; a priori estimates; weak solutions; kinetic energy; dissipation

---

## 1. Introduction

During the last century, the Navier–Stokes equations attracted very much attention. The first essential steps in this way were offered by C. Oseen [1], F. K. G. Oldquist [2], J. Leray [3–5], and E. Hopf [6]. Later, the Cauchy problem and the boundary value problem were actively studied by many authors (see, for example, [7,8], the review [9–17] and etc.). The main objects and tools of these works were weak solutions or fix points of integral operators. Here, a special case is connected with the existence problem of a global and regular solution in the 3D Cauchy problem. In response to the new setting of this task by Ch. Fefferman in 2000 (see [18]), O.A. Ladyzhenskaya wrote in her review [9] that she would put the main question otherwise: "Do or don't the Navier–Stokes equations give, together with initial and boundary dates, the deterministic description of fluid dynamics?"

Then, this problem is more difficult and more interesting from the physical point of view. Therefore, I introduced some invariants for studying solutions properties. At least, it is natural for applications because invariants are very important and strong tools. Moreover, these invariants didn't apply earlier.

Let us describe them now. The first invariant connected with the Cauchy problem that provided initial data belongs to a special class $C_{6/5,\,3/2}^\infty$ of solenoidal vector fields vanishing at infinity. Here, outer forces are trivial. Then, the class $C_{6/5,\,3/2}^\infty$ is invariant (Theorem 2). This is a new result.

The second invariant is a special parameter $\lambda$ (see (68)) which is connected with a velocity changing of $E^2$, where $E$ is a kinetic energy of a fluid flow. If $\lambda \geq 1$ or kinetic energy at a special moment is not less any mean depending on $\lambda$ for $\lambda < 1$ ( i.e., changing of $E^2$ at moment $t = 0$ is negligible), then an ideal, global and smooth motion is determined. In other words, a global regular solution exists (Theorem 7). This is an essential and qualitative improvement of the classical result together with a new a priori estimate given by Theorems 8–10. These theorems are new results in principle.

Finally, the other parameters $\varepsilon, 0 < \varepsilon < 1$, (see formula (87)) and $\mu, 1 < \mu < \lambda^{-4}$, or $\mu = \infty$ may also be very useful (see formula (69), Lemma 50). The first of them is a dissipation coefficient of kinetic energy . The last parameter holds a time interval of a solution regularity. These three numerical characteristics $\lambda, \varepsilon, \mu$ are invariant with respect to the scaling procedure.

By the way, the first attempts to estimate invariant norms were implicitly undertaken in [12,16].

An introduction of a special invariant class of vector fields and invariant parameters gives the main idea for the proof of basic results. The first step is connected with a change of the construction offered in [19]. These changes concern solution approximations. The special kind of them gives many uniform a priori estimates. Approximations of a velocity function are built on a fundamental system with a condition for Laplacians of approximative solutions. They must be a finite part of the Fourier series. Simultaneously, approximations of a pressure function are being built. Jointly with a hydrodynamical potential, these approximations give the following facts and properties of local solutions:

(1)　solutions are bounded with respect to a uniform norm and therefore it belongs to any class $L_{p,\,q}$;
(2)　there is a universal time interval $[0, T_0)$ where bounded solutions exist;
(3)　more exact necessary conditions of a hypothetical turbulence phenomenon if it is;
(4)　a lower estimate of the kinetic energy which influences an existence of a global smooth solution.

The last two items are very important. If dissipation of kinetic energy is large (close to the unit), then blow up is probable.

To the structure of the paper. In the first part (Section 2), there are considered solutions' properties of the Cauchy problem in a local form if initial data is smooth enough. Here, there is given a modification of classical results with some supplements (see Theorem 1).The rest of this part contains technical lemmas which are proved by application of hydrodynamic potentials and multiplicative inequalities from Appendix (Appendix A). In the second part (Sections 3 and 4), there are existence conditions of global solutions studied in this problem, conditions for local solutions' extensions if the kinetic energy is small and close to the minimum. A more precise hypothetic blow up time interval is found. Here, three basic parameters $\lambda, \mu, \varepsilon$ are very useful.

The third part (Sections 5 and 6) contains the proof of main statements (Theorems 7–9), which are based on properties of invariant parameters $\lambda, \mu, \varepsilon$.

I think, in this way, it is convenient to remove any restrictions on a smoothness in some contrast to the traditional way. The main idea is connected with an invariant form of an a priori estimate for gradient norms of a velocity. In addition, other norms are estimated in class $L_6$ and, after that, it is done in class $L_2$. In particular, it is shown that there is a bad solution of a class $L_6$ with some good properties. As the corollary, this solution has many uniformly bounded norms with respect to time argument. Only after that, by routine calculations, we prove the bad solution from above belongs to a class $L_2$. Precisely, this step distinguishes from classical way for the second time (see [7]).

In the considered problem, a boundedness of solutions depends on a smoothness of initial data. At least, initial data from the Sobolev class $W_2^3$ gives the same in principle.

The offered construction doesn't permit diminishing the index of smoothness.

In the final (Section 7), we explain the principal difference between the Navier–Stokes equations in space and plane.

A part of local results in modification (Section 2) and invariants as tools (Section 4) were announced by author in [20–22].

NOTATION. Now, let us consider the Cauchy problem ($n = 3$):

$$D_t u_k + \sum_{i=1}^{n} u_i u_{k,\,i} = \nu \Delta u_k - P_{,\,k}, \ k = 1, 2, \ldots, n, \tag{1}$$

$$divu = 0, \ u(0, x) = \varphi(x), \tag{2}$$

where $u$ is a velocity of flow, $P$ is a pressure function, symbols

$$D_t u = \frac{\partial u}{\partial t}, u_{k,i} = \frac{\partial u_k}{\partial x_i}, \ u_{k,ij} = \frac{\partial^2 u_k}{\partial x_i \partial x_j}, ..., P_{,k} = \frac{\partial P}{\partial x_k}$$

indicate a partial differentiation or differentiation in distributions, $\triangle$ is the Laplace operator, and $\nu$ is a positive constant (viscosity coefficient). A mapping $\varphi$ has all derivatives and satisfies conditions of averaged growth: $\varphi \in L_{6/5}(R^3)$, $\varphi_{,i} \in L_{3/2}(R^3)$. The other derivatives belong to classes $L_r(R^3)$ for any $r > 1$. Furthermore, this class is denoted by symbol $C_{6/5,\,3/2}^\infty$. A class $C_0^\infty(R^n)$ is the class of infinitely smooth mappings with a compact support. A norm in a space $L_p(\Omega)$ is defined by formula:

$$\|v\|_p = \left( \int_\Omega |v(x)|^p dx \right)^{1/p}.$$

A mixed norm is defined by equality:

$$\|u\|_{p,\,q} = \left( \int_0^T \left( \int_\Omega |u(t,x)|^p dx \right)^{\frac{q}{p}} dt \right)^{\frac{1}{q}}.$$

A symbol $D^\alpha v$ denotes a partial differentiation or distributions with respect to a multi–index $\alpha$. An order of the derivative is indicated by $|\alpha|$. Jacobi matrix of a mapping $v$ with respect to spatial variables is denoted by $\nabla v$. Its modulus is

$$|\nabla v| = \left( \sum_{i,\,j} v_{i,\,j}^2 \right)^{\frac{1}{2}}. \tag{3}$$

Functions' properties from the Sobolev classes $W_p^l(\Omega)$ are given, for example, in [23–25]. A norm in this functional space is defined by

$$\|v\|_{W_p^l(\Omega)} = \sum_{|\alpha| \le l} \|D^\alpha v\|_p.$$

Let $v$ be a mapping that is determined on the whole space. For the Riesz potential, we apply notation:

$$I_\alpha(v)(x) = \frac{1}{\gamma(\alpha)} \int_{R^n} \frac{v(y)dy}{|x-y|^{n-\alpha}}, \tag{4}$$

where $\gamma(\alpha) = \pi^{\frac{n}{2}} \Gamma(\frac{\alpha}{2}) / \Gamma(\frac{n-\alpha}{2})$ and $\Gamma$ is the Euler gamma–function. The properties of these potentials can be found in [24].

*The agreement about summation.* Everywhere in this article, the repeated indices give a summation if it is not done reservation specially. For example,

$$u_i u_{j,i} = \sum_{i=1}^n u_i u_{j,i}, \ u_{i,j} u_{j,i} = \sum_{i,j=1}^n u_{i,j} u_{j,i}, \ u_i u_{j,i} \triangle u_j = \sum_{i,j=1}^n u_i u_{j,i} \triangle u_j,$$

etc.

Furthermore, $S_T = [0,T] \times R^3$. A number $T_0$ we define by formula:

$$T_0 = \left( \frac{9}{4} \right)^4 \frac{\nu^3}{\|\nabla \varphi\|_2^4}. \tag{5}$$

We apply the definition of a weak solution given in [7] everywhere.

## 2. Preliminaries. Boundedness and Smoothness Properties of Local Solutions in the Cauchy Problem

Here, with some compliments, a local result described by Theorem 1 is basic in this section. The rest contains only technical statements.

**Theorem 1.** *Let $T_0$ be a number from formula (5) and a mapping $\varphi \in C^\infty_{6/5,\, 3/2}$. Then, on the set $S_{T_0}$, there exist weak solutions u and P of problems (1) and (2) with the following properties:*

(1)  *mappings u and P uniformly continuous and bounded on the set $S_T$ for every number T, $0 < T < T_0$;*

(2)  *the solution u belongs to Sobolev classes $W_2^2(S_T)$ and $W_6^1(S_T)$ for every number T, $0 < T < T_0$, moreover, all norms*
$$\|u\|_p, \ \|\nabla u\|_p, \ \|D_t u\|_p, \ \|u_{,ij}\|_p, \ \|\nabla D_t u\|_2$$
*are uniformly bounded in spaces $L_p(R^3)$, $2 \le p \le 6$, by a constant $C = C(\nu, \varphi, T)$ depending on $\nu$, $\varphi$ and T only, in addition $\|u\|_2 \le \|\varphi\|_2$;*

(3)  *gradients $\nabla u_i$, $i = 1, 2, 3$, $\nabla P$ are bounded on the set $S_T$ for every number T, $0 < T < T_0$;*

(4)  *the solution P satisfies uniform estimates:*
$$\|\nabla P\|_q \le C, \ \frac{3}{2} < q < \infty, \ \|\nabla D_t P\|_q \le C, \ \|P_{,ij}\|_q \le C,$$
*for all numbers q, $\frac{3}{2} < q \le 3$, and $t \in [0, T]$, $T < T_0$, with constants C depending on $\nu$, $\varphi$, T and q only;*

(5)  *solutions u and P are classical solutions that is for any $T < T_0$ they belong to the class $C^\infty((0, T_0) \times R^3) \bigcap C(S_T)$.*

The proof of the theorem is given to the end of this section. We note items 1, 3, 4 compliment well-known Ladyzhenskaya's results (see [7]). Item (2) contains new uniform estimate for norms of derivatives. Hence, it follows a boundedness of weak solutions and a finiteness of its mixed norms. Moreover, we have an existence of weak solution with required properties on the interval $[0, T_0)$ with the finite length. To the studying of the smoothness property for weak solutions, the mixed norms were applied by O. Ladyzhenskaya in [26] (see, also [7]). They were applied by other authors (see, for example, [8,10,14]). Item (5) is a particular case from [27]. However, from this theorem, a deeper result follows (see Theorem 7).

### 2.1. A Priori Estimates of Gradients' Norms

**Lemma 1.** *Suppose that a mapping $w : S_{T_0} \to R^3$ belongs to a class $C^2$ and $w(0, x) = \varphi(x)$. If, for every $t \in [0, T_0)$, Laplacian supports are subsets of some ball with a fixed radius and $w \in L_6(R^3)$, $\nabla w \in L_2(R^3) \bigcap L_6(R^3)$, then for all mappings w satisfying condition:*

$$\frac{1}{2}\frac{d}{dt}\|\nabla w\|_2^2 + \nu\|\triangle w\|_2^2 = \int_{R^3} w_i w_{k,\,i} \triangle w_k dy, \tag{6}$$

*the following estimate holds:*

$$\|\nabla w\|_2 \le \frac{\|\nabla \varphi\|_2}{\left(1 - t/T_0\right)^{1/4}}$$

*for all $t \in [0, T_0)$, where $T_0$ from formula (5).*

**Proof.** We take from Corollary A4 the second inequality. Then, from (6), we obtain:

$$\frac{1}{2}\frac{d}{dt}\|\nabla w\|_2^2 + \nu\|\triangle w\|_2^2 \le a_1\|\nabla w\|_2^{3/2}\|\triangle w\|_2^{3/2}. \tag{7}$$

Let $y = \|\triangle w\|_2 / \|\nabla w\|_2^3$. Then, (7) can be rewritten in the form:

$$\frac{1}{2\|\nabla w\|_2^6} \frac{d}{dt} \|\nabla w\|_2^2 \leq a_1 y^{3/2} - \nu y^2.$$

The maximal mean on the right-hand side is $\frac{27a_1^4}{256\nu^3}$. Therefore, integrating the inequality

$$\frac{1}{2\|\nabla w\|_2^6} \frac{d}{dt} \|\nabla w\|_2^2 \leq \frac{27a_1^4}{256\nu^3}$$

over the interval $[0, t]$, we get:

$$\frac{1}{\|\nabla \varphi\|_2^4} - \frac{1}{\|\nabla w\|_2^4} \leq \frac{27a_1^4}{64\nu^3} t.$$

Furthermore, we take a number $a_1$ from Corollary A4 and obtain the required estimate. $\quad\square$

**Lemma 2.** *Let $T_0$ be a constant from Lemma 1. Assume a mapping $w : S_{T_0} \to R^3$ belongs to a class $C^3$ and $w(0, x) = \varphi(x)$, $D_t w(0, x) = \psi(x)$. Suppose that, for every $t$, there are fulfilled conditions:*

*(1)    Laplacian supports $\triangle w$, $\triangle D_t w$ are subsets of a ball with a fixed radius;*

*(2)    mappings*

$$w, \ D_t w \in L_6(R^3), \ \nabla w, \ \nabla D_t w \in L_2(R^3) \bigcap L_6(R^3);$$

*(3)    with constants $k_1$, $l$ the inequalities hold:*

$$\|\nabla w\|_2 \leq k_1 \|\nabla \varphi\|_2, \ \int_0^t \|\triangle w\|_2^2 dt \leq l;$$

*(4)    the equality*

$$\frac{1}{2} \frac{d}{dt} \|\nabla D_t w\|_2^2 + \nu \|\triangle D_t w\|_2^2 = \int_{R^3} \left( D_t w_i \cdot w_{k,\, i} + w_i D_t w_{k,\, i} \right) \triangle D_t w_k dx \qquad (8)$$

*is true. Then, for every segment, $[0, T]$ where $T < T_0$ the estimate $\|\nabla D_t w\|_2 \leq k_2 \|\nabla \psi\|_2$ holds with a constant $k_2$ which depends on $\nu, T, k_1, l, \|\nabla \varphi\|_2$ only.*

**Proof.** The integral on the right-hand side in formula (8) we rewrite with two integrals $J_1$ and $J_2$. Applying Corollary A4, we make estimates for every integral. In integral $J_1$, a triple of mappings $u, v, w$ is the triple $D_t w, \ w, \ D_t w$. In integral $J_2$, a required triple is the triple $w, \ D_t w, \ D_t w$. Therefore, condition (3) yields estimates:

$$J_1 \leq a \|\nabla D_t w\|_2 \|\nabla w\|_2^{1/2} \|\triangle w\|_2^{1/2} \|\triangle D_t w\|_2$$

$$\leq a \sqrt{k_1} \|\nabla \varphi\|_2^{1/2} \|\nabla D_t w\|_2 \|\triangle w\|_2^{1/2} \|\triangle D_t w\|_2,$$

$$J_2 \leq a \|\nabla w\|_2 \|\nabla w\|_2^{1/2} \|\triangle w\|_2^{1/2} \| \leq ak_1 \|\nabla \varphi\|_2 \|\nabla D_t w\|_2^{1/2} \|\triangle w\|_2^{3/2}.$$

Hence, from (8), we get:

$$\frac{1}{2} \frac{d}{dt} \|\nabla D_t w\|_2^2 + \nu \|\triangle D_t w\|_2^2 \leq \qquad (9)$$

$$a \sqrt{k_1} \|\nabla \varphi\|_2^{1/2} \|\nabla D_t w\|_2^{1/2} \|\triangle D_t w\|_2 \left( \|\triangle w\|_2^{1/2} + \sqrt{k_1} \|\nabla \varphi\|_2^{1/2} \|\triangle D_t w\|_2^{1/2} \right).$$

Let $g(t) = \|\nabla D_t w\|_2$, $h(t) = \|\triangle D_t w\|_2 / g(t)$. Then, formula (9) can be transformed to the formula:

$$\frac{1}{2} \frac{d}{dt} \ln g(t) + \nu h^2(t) \leq a \sqrt{k_1} \|\nabla \varphi\|_2^{1/2} \|\triangle w\|_2^{1/2} h(t) + ak_1 \|\nabla \varphi\|_2 h^{3/2}(t).$$

Let us integrate over segment $[0, t]$ this inequality. For the next step, we apply to each term the Hölder inequality for three and two factors, respectively getting quantities $h^2(t)$ and $\|\triangle w\|_2^2$. Hence, from condition (3), we obtain:

$$\frac{1}{2}\ln\frac{g(t)}{g(0)} + \nu\int_0^t h^2(t)dt \leq a\sqrt{k_1\|\nabla\varphi\|_2}\sqrt[4]{t}\left(\int_0^t\|\triangle w\|_2^2 dt\right)^{1/4}\left(\int_0^t h^2(t)dt\right)^{1/2} +$$

$$+ak_1\|\nabla\varphi\|_2\sqrt[4]{t}\left(\int_0^t h^2(t)dt\right)^{3/4} \leq a\sqrt{k_1\|\nabla\varphi\|_2}\sqrt[4]{lt}\sqrt{y} + ak_1\|\nabla\varphi\|_2\sqrt[4]{t}y^{3/4},$$

where $y = \int_0^t h^2(t)dt$. Let $M$ be a maximal mean of the function

$$F(y) = a_1\sqrt{y} + a_2 y^{3/4} - \nu y,$$

where $a_1 = a\sqrt{k_1\|\nabla\varphi\|_2}\sqrt[4]{lt}$, $a_2 = ak_1\|\nabla\varphi\|_2\sqrt[4]{t}$. Then, the last estimates give $g(t) \leq e^{2M}g(0)$. From the definition of function $g$, we have $g(0) = \|\nabla\psi\|_2$. $\square$

*2.2. A Priori Estimates of Laplacian Norms*

**Lemma 3.** *Let $w$, $T_0$ be a mapping and a number from Lemma 1. Then, for every number $T$, $0 < T < T_0$, there exists a constant $l = l(\nu, \varphi, T)$ such that*

$$\int_0^t\|\triangle w\|_2^2 dt \leq l$$

*for all $t \in [0, T]$.*

**Proof.** We transform inequality (7) applying the estimate from Lemma 1. Then,

$$\frac{1}{2}\frac{d}{dt}\|\nabla w\|_2^2 + \nu\|\triangle w\|_2^2 \leq a\frac{\|\nabla\varphi\|_2^{3/2}}{(1 - t/T_0)^{3/8}}\|\triangle w\|_2^{3/2}.$$

This inequality we integrate over the segment $[0, t]$. Then, we estimate the right-hand side applying the Hölder inequality and underlining the integral with the term $\|\triangle w\|_2^2$. If $\beta(t) = \int_0^t\|\triangle w\|_2^2 dt$, then we get

$$\frac{1}{2}\|\nabla w\|_2^2\Big|_0^t + \nu\beta(t) \leq a\|\nabla\varphi\|_2^{3/2}\beta^{3/4}(t)\left(\int_0^t(1 - t/T_0)^{-3/2}dt\right)^{1/4}.$$

The direct calculations of the integral on the right-hand side and the estimate

$$\frac{1}{\sqrt{1-b}} - 1 \leq \frac{b}{\sqrt{1-b}}$$

give the inequality:

$$\frac{1}{2}\|\nabla w\|_2^2 + \nu\beta(t) \leq a\|\nabla\varphi\|_2^{3/2}\beta^{3/4}(t)\frac{\sqrt[4]{2t}}{(1 - t/T_0)^{1/8}} + \frac{1}{2}\|\nabla\varphi\|_2^2.$$

Take out the first term on the left hand. Then, the required estimate for function $\beta(t)$ will be obvious. If $\beta(t) \leq \|\nabla\varphi\|_2$, then the estimate is acceptable. If $\beta(t) \geq \|\nabla\varphi\|_2$, then we have:

$$\nu\beta(t) \leq a\|\nabla\varphi\|_2^{3/2}\beta^{3/4}(t)\frac{\sqrt[4]{2t}}{(1 - t/T_0)^{1/8}} + \frac{1}{2}\|\nabla\varphi\|_2^{5/4}\beta^{3/4}(t).$$

Hence, it follows the lemma. $\square$

**Lemma 4.** *Let w be a mapping from Lemma 2 and a number $T_0$ from Lemma 1. Then, for every number T,*
*$0 < T < T_0$, there exists a constant $l_1 = l_1(v, \varphi, T)$ such that*

$$\int_0^t \|\triangle D_t w\|_2^2 dt \leq l_1$$

*for all $t \in [0, T]$.*

**Proof.** For the mapping $w$, inequality (9) is fulfilled. Its right-hand side we estimate relying on Lemma 2. Then,

$$\frac{1}{2}\frac{d}{dt}\|\nabla D_t w\|_2^2 + v\|\triangle D_t w\|_2^2 \leq \tag{10}$$

$$a\sqrt{k_1}k_2\|\nabla\varphi\|_2^{1/2}\|\nabla\psi\|_2\|\triangle w\|_2^{1/2}\|\triangle D_t w\|_2 + ak_1\sqrt{k_2}\|\nabla\varphi\|_2^{1/2}\|\nabla\psi\|_2\|\triangle D_t w\|_2^{3/2}.$$

Let $C$ be a maximal coefficient of factors

$$\|\triangle w\|_2^{1/2}\|\triangle D_t w\|_2, \ \|\triangle D_t w\|_2^{3/2}.$$

Therefore, from formula (10), we have inequality:

$$\frac{1}{2}\frac{d}{dt}\|\nabla D_t w\|_2^2 + v\|\triangle D_t w\|_2^2 \leq C\|\triangle w\|_2^{1/2}\|\triangle D_t w\|_2 + C\|\triangle D_t w\|_2^{3/2}.$$

This inequality we integrate over segment$[0, t]$ and its right-hand side we estimate applying the Hölder inequality and underlining terms with norms $\|\triangle D_t w\|_2$. If

$$\beta_1(t) = \int_0^t \|\triangle, D_t w\|_2^2 dt$$

then we have the estimate:

$$\frac{1}{2}\|\nabla D_t w\|_2^2\Big|_0^t + v\beta_1(t) \leq C\sqrt[4]{t}\Big(\int_0^t \|\triangle w\|_2^2 dt\Big)^{1/4}\beta_1^{1/2}(t) + C\sqrt[4]{t}\beta_1^{3/4}(t).$$

We increase the right side using Lemma 3 and deduce the left side taking out the first positive term. Then, we obtain:

$$v\beta_1(t) \leq C\sqrt[4]{lt}\beta_1^{1/2}(t) + C\sqrt[4]{t}\beta_1^{3/4}(t) + \frac{1}{2}\|\nabla\psi\|_2^2.$$

Hence, we get the lemma in the same way as Lemma 3. If $\beta_1(t) > \|\nabla\psi\|_2$, then, from

$$\|\nabla\psi\|_2^2 < \beta_1^{1/2}(t)\|\nabla\psi\|_2^{3/2},$$

we obtain the lemma inequality. If $\beta_1(t) \leq \|\nabla\psi\|_2$, then the estimate is acceptable. □

*2.3. Basic Space of Solenoidal Vector Fields and Orthogonal Systems*

Let us consider solenoidal vector fields $\varphi : R^3 \to R^3$ from class $C^\infty$ with a compact support of $\triangle\varphi$. A closure of this class is defined by the norm:

$$\|\varphi\| = \|\varphi\|_6 + |\nabla\varphi\|_2 + \sum_{i,j}\|\varphi, ij\|_2. \tag{11}$$

We denote its by $J_0^2(R^3)$. From Lemmas A1 and A2, it follows that elements $u \in J_0^2(R^3)$ are represented by the Riesz potentials; moreover, $u, \nabla u \in L_6(R^3)$. Otherwise, each element is defined uniquely by its Laplacian. The class $J_0^2(R^3)$ is a separable space as a subspace of the Sobolev classes

$W_p^l(R^3), 1 < p < \infty$. Therefore, there exists a countable system $(\psi^n)_{n=1,\dots}$ of infinite smooth vector fields satisfying conditions:

(1)  $div\ \psi^n = 0$;
(2)  supports of $\triangle\psi^n$ are compact sets;
(3)  the closure of a linear span in norm (11) coincides with the space $J_0^2(R^3)$.

Now, we apply the Sonin–Shmidt orthogonalization to the fundamental system $(\psi^n)_{n=1,\dots}$ and construct a countable system of mappings $(b^n)_{n=1,\dots}$, which would be with the orthogonality property of Laplacians in the space $L_2(R^3)$. That is, the scalar product

$$(\triangle b^n, \triangle b^m) = \int_{R^3} \triangle b_i^n \triangle b_i^m dx = \delta_{ij}, \tag{12}$$

where $\delta_{ij}$ is Kronecker's symbol. Then, every mapping $b^n$ is a finite linear combination of mappings $(\psi^k)$. Therefore, a support of $\triangle b^n$ is a compact set. Let

$$\triangle b^n = a^n. \tag{13}$$

The system $(a^n)$ is complete for the space $J_0^2(R^3)$; that is, the following proposition is true.

**Lemma 5.** *If in the space $L_2(R^3)$ for a some vector field $u \in J_0^2(R^3)$ the scalar product $(u, a^n) = 0$ for every $n = 1, 2, \dots$ then $u = 0$.*

**Proof.** From chosen mappings $a^n$, the equality $(u, \triangle\psi^n) = 0$ for each element of the fundamental system $(\psi^n)_{n=1,\dots}$ follows. The Stokes theorem gives

$$\int_{|x|\leq r} u_k \triangle\psi_k^n dx = -\int_{|x|\leq r} u_{k,i}\psi_{k,i}^n dx + \int_{|x|=r} u_k \psi_{k,i}^n \frac{x_i}{r} dS.$$

The integral over the sphere vanishes as $r \to \infty$. Actually, from Corollary A2 of Lemma A2, we have:

$$\left| \int_{|x|=r} u_k \psi_{k,i}^n \frac{x_i}{r} dS \right| \leq \frac{C_1}{r^2} \int_{|x|=r} |u(x)| dS.$$

Furthermore, we apply Lemma A4 ($\alpha = 2,\ p = 6$) taking into consideration a continuity of $u$. The passage to the limit yields the equality $(u, \triangle\psi^n) = -(\nabla u, \nabla\psi^n)$ or $(\nabla u, \nabla\psi^n) = 0$. We take a sequence of finite and smooth mappings

$$(\eta^n)_{n=1,\dots},\ \eta^n \in J_0^2(R^3),\ \eta^n = \sum_i^n \beta_i \psi^i,$$

which converges to the vector field $u$ in the space $J_0^2(R^3)$. Hence, $(\nabla u, \nabla u) = 0$. The summability of $u$ in the space $L_6(R^3)$ proves lemma equality. $\square$

**Remark 1.** *To the fundamental system of mappings $(\psi^n)$ we can adjoin any solenoidal vector field $\varphi \in C_0^\infty(R^3), \varphi \neq 0$ or any vector field from the class $J_0^2(R^3)$ as the first element of this system.*

*2.4. Successive Approximations of Solutions and Its Estimates: Velocity*

Let $(a_{1,\dots}^n)$ be an orthonormal system of mappings in the space $L_2(R^3)$ constructed above with the completeness property in $J_0^2(R^3)$ and conditions (12) and (13). Moreover, $a^n \in C_0^\infty(R^3)$ for all $n$ and $a^1 = \frac{\triangle\varphi}{\|\triangle\varphi\|_2}$ where a vector field $\varphi \in C_0^\infty(R^3)$ is initial data in problems (1) and (2).

For successive approximations $v^n$, we define changing Ladyzhenskaya's construction in ([7], p. 197). Set

$$\triangle v^n(t, x) = \sum_{q=1}^{n} c_{qn}(t) a^q(x). \tag{14}$$

Then, an approximative solution $v^n$ is built as a hydrodynamical potential

$$v^n(t, x) = -\frac{1}{4\pi} \sum_{q=1}^{n} c_{qn}(t) \int_{R^3} \frac{a^q(y) dy}{|x - y|}. \tag{15}$$

Functions $c_{qn}$ are solutions of a system of differential equations:

$$\left( D_t v^n, a^q \right) - \nu \left( \triangle v^n, a^q \right) + \int_{R^3} v_i^n v_{k,i}^n a_k^q dx = 0, \ q = 1, 2 \dots, n, \tag{16}$$

with initial data: $c_{qn}(0) = \|\triangle \varphi\|_2 \delta_{q1}$, where $\delta_{qr}$ is Kronecker's delta. Hence,

$$\triangle v^n(0, x) = \triangle \varphi(x), \ v^n(0, x) = \varphi(x). \tag{17}$$

Now, we find an existence interval of a smooth solution in system (16). For every equation from (16), we multiply by functions $c_{qn}$ and sum them. As a result, we have:

$$\left( D_t v^n, \triangle v^n \right) - \nu \|\triangle v^n\|_2^2 + \int_{R^3} v_i^n v_{k,i}^n \triangle v_k^n dx = 0.$$

From Corollary A3, we get:

$$\left( D_t \nabla v^n, \nabla v^n \right) + \nu \|\triangle v^n\|_2^2 = \int_{R^3} v_i^n v_{k,i}^n \triangle v_k^n dx. \tag{18}$$

From Lemmas A1 and A2 vector fields $v^n \in L_6(R^3)$, $\nabla v^n \in L_6(R^3) \bigcap L_2(R^3)$. Equalities (17) and (18) are conditions of Lemma 1 for mappings $v^n$. Therefore, in system (16), an existence of smooth solutions on some interval $[0, t_0)$ is guaranteed by well-known theorems for ordinary differential equations. By Lemma 1 (see estimates), these solutions can be extended on the interval $[0, T_0)$ where $T_0$ is the constant in Lemma 1 (see also (5)). Thus, we proved the following statement.

**Lemma 6.** *Let $[0, T_0)$ be an interval from Lemma 1. Then, for every $t \in [0, T_0)$, approximations $v^n$ constructed by formulas (14) and (16) satisfy conditions:*

*(1)* $\|\nabla v^n\|_2 \leq \|\nabla \varphi\|_2 \left( 1 - t/T_0 \right)^{-1/4}$,

*(2)* $\|\nabla v^n\|_6 \leq A\|\triangle v^n\|_2$,

*where a constant A from Lemma A1.*

**Proof.** Item (1) follows from Lemma 1. Item (2) is the corollary of the second representation in (A1), Lemma A1 and arguments in the proof of Corollary A4. □

**Lemma 7.** *Let $[0, T_0)$ be a constant of Lemma 1. Then, for every segment $[0, T]$, $T < T_0$, approximations $v^n$, which are constructed by formulae (14)–(16), satisfy inequalities:*

*(1)* $\|\nabla D_t v^n\|_2 \leq k_2 \|\nabla (\nu \triangle \varphi - \varphi_i \varphi_{,i})\|_2$, *where a number $k_2 = k_2(\nu, \varphi, T)$ depends on $\nu, \varphi, T$ only;*

*(2)* $\|\nabla D_t v^n\|_6 \leq A\|\triangle D_t v^n\|_2$ *with the constant A from Lemma A1.*

**Proof.** The item (2) can be proved in the same way as the estimate (2) from Lemma 6. Let us prove item (1). We differentiate equalities (16) with respect to $t$. Then, from each, we multiply by the derivative $c'_{qn}(t)$ and add together in final. As a result, we have

$$\left( D_{tt}v^n, \triangle D_t v^n \right) - \nu \| \triangle D_t v^n \|_2^2 + \int_{R^3} \left( D_t v_i^n v_{k,\,i}^n + v_i^n D_t v_{k,\,i}^n \right) \triangle D_t v_k^n dx = 0.$$

A support of $\triangle D_t v^n$ is a compact set. The Stokes theorem and Corollary A3 give:

$$\left( D_{tt} \nabla v^n, \nabla D_t v^n \right) + \nu \| \triangle D_t v^n \|_2^2 = \int_{R^3} \left( D_t v_i^n v_{k,\,i}^n + v_i^n D_t v_{k,\,i}^n \right) \triangle D_t v_k^n dx. \tag{19}$$

From Lemmas A1 and A2, we have

$$v^n, D_t v^n \in L_6(R^3), \ \nabla v^n, \ D_t \nabla v^n \in L_6(R^3) \bigcap L_2(R^3).$$

By Lemma 3, the vector field $v^n$ satisfies the inequality:

$$\int_0^t \| \triangle v^n \|_2^2 dt \le l(\nu, \varphi, T).$$

Then, mappings $v^n$ satisfy Lemma 2. This implies:

$$\| \nabla D_t v^n \|_2 \le k_2 \| \nabla D_t v^n(0, x) \|_2 \tag{20}$$

with some constant $k_2 = k_2(\nu, \varphi, T)$.

Let us estimate the right-hand side of (20). In (16), we take $t = 0$. Then, we multiply them by numbers $c'_{qn}(0)$ respectively and add them together. As a result, formula (17) gives

$$\left( D_t v^n(0, x), \triangle D_t v^n(0, x) \right) - \left( \triangle D_t v^n(0, x), T^1(x) \right) = 0, \tag{21}$$

where

$$T^1 = \nu \triangle \varphi - \varphi_i \varphi_{,\,i}. \tag{22}$$

We move derivatives with the factor $\triangle D_t v^n$ in (21) using Corollary A3 and a finiteness of mapping $\varphi$. Then, we obtain

$$\| \nabla D_t v^n(0, x) \|_2^2 = \left( \nabla D_t v^n(0, x), \nabla T^1(x) \right).$$

Apply Cauchy–Bunyakovskii's inequality. Hence, we get the required estimate

$$\| \nabla D_t v^n(0, x) \|_2 \le \| \nabla T^1(x) \|_2.$$

Thus, from (20), a lemma follows. □

**Lemma 8.** *Let $T_0$ be a constant of Lemma 1. Then, on every segment $[0, T]$, $0 < T < T_0$, approximations $v^n$ from formulae (14)–(16) satisfy conditions:*

*(1)　$\| \triangle v^n \|_2 \le C = C(\nu, \varphi, T)$;*
*(2)　$\int_0^t \| \triangle D_t v^n \|_2^2 dt \le l_1$;*

*where constants $C$ and $l_1$ depend on $\nu, \varphi, T$ only.*

**Proof.** Condition (1) follows from (18). We apply the Cauchy–Bunyakovskii inequality and estimates (1) of Lemmas 6 and 7 to the scalar product $\left( D_t \nabla v^n, \nabla v^n \right)$. Then, $\left| \left( D_t \nabla v^n, \nabla v^n \right) \right| \le C_1(\nu, \varphi, T) = C_1$. The right-hand side from (18) is estimated by applying Corollary A4, where we take the triple $v_n, v_n, v_n$. From (18), we have

$$\nu \| \triangle v^n \|_2^2 \le C_1 + a \| \nabla v^n \|_2^{3/2} \| \triangle v^n \|_2^{3/2}.$$

Apply again estimate (1) of Lemma 6. Then,

$$\nu \|\triangle v^n\|_2^2 \le C_1 + C_2 \|\triangle v^n\|_2^{3/2}.$$

This implies condition (1). Vector fields $v^n$ satisfy Lemma 2 (see the proof of Lemma 6). Then, Lemma 4 gives estimate (2). □

**Lemma 9.** *Let $T_0$ be a constant of Lemma 1. Then, approximations $v^n$ from (14)–(16) are bounded by a constant $C$ on the set $S_T$ for every $T < T_0$ where a constant $C$ depends on $\nu, \varphi, T$ only.*

**Proof.** For approximation $v^n$, we use integral representation (A2). One should replace integration over whole space by integrations over ball $|y - x| \le 1$ and its complement. Then, $v^n(t, x) = \frac{1}{4\pi}(J_1 + J_2)$. Every term is estimated by application of Hölder's inequality. We have

$$|J_1| \le \|\nabla v^n\|_6 \Big( \int_{|y-x| \le 1} \frac{dy}{|x-y|^{2,4}} \Big)^{5/6}, |J_2| \le \|\nabla v^n\|_2 \Big( \int_{|y-x| \ge 1} \frac{dy}{|x-y|^4} \Big)^{1/2}.$$

Hence,

$$|J_1| \le C_1 |\nabla v^n|_6, \ |J_2| \le C_2 |\nabla v^n|_2,$$

where $C_1, C_2$ are universal constants. The norm $\|\nabla v^n\|_6$ is estimated in two steps. In the first step, we apply inequality 2) from Lemma 6. After that, we use inequality (1) from lemma 8. To estimate another norm $\|\nabla v^n\|_2$, we can apply inequality (1) from Lemma 6. Hence, we get a boundedness of all vector fields $v^n$ by a general constant. □

**Lemma 10.** *Let $T_0$ be a constant of Lemma 1. Then, for every exponent $p \in [3/2, 6]$ and every segment $[0, T]$, $T < T_0$, approximations $v^n$ from formulae (14)–(16) satisfy the inequality $\|v_i^n v_{,i}^n\|_p \le C$ with a constant $C$ depending on $\nu, \varphi, T, p$ only.*

**Proof.** If $p = 6$, then the statement follows from Lemma 9 and estimates by item (2) of Lemma 6 and item (1) of Lemma 8. If $p = 3/2$, then we apply Hölder's inequality. Hence, we have $\|v_i^n v_{,i}^n\|_{3/2} \le \|v^n\|_6 \|\nabla v^n\|_2$. Estimates of Lemma 6 and Lemma 8 prove the lemma for this exponent. An intermediate exponents is verified by Lemma A5. □

**Lemma 11.** *Let $T_0$ be a constant of Lemma 1. Then, for all $t \in [0, T_0)$, approximations $v^n$ from formulae (18)–(20) satisfy inequalities*

$$\|v_{,ij}^n\|_2 \le M |\triangle v^n\|_2, \ \|D_t v_{,ij}^n\|_2 \le M |\triangle D_t v^n\|_2,$$

$i, j = 1, 2, 3$, *with a universal constant $M$.*

**Proof.** The statement of lemma is the corollary well-known results about integral differentiation with a weak singularity (see [28]). From the second representation of Lemma A2, we obtain two equalities: $v_{,ij}^n = k_{ij} \triangle v^n + T_{ij}(\triangle v^n)$, $D_t v_{,ij}^n = k_{ij} \triangle D_t v^n + T_{ij}(\triangle D_t v^n)$, where $k_{ij}$ are some constants, $T_{ij}$ are singular integral operators. Its boundedness in the space $L_2$ gives the required estimates. □

**Lemma 12.** *Let $T_0$ be a constant of Lemma 1. Then, for every segment $[0, T]$, $T < T_0$, for all exponents $p \in [1, 3/2]$ and each triple $i, j, k = 1, 2, 3$ approximations $v^n$ from (14)–(16) satisfy inequalities:*

(1) $\|v_{i,j}^n v_{j,ik}^n\|_p \le C$;

(2) $\|v_{i,j}^n D_t v_{j,ik}^n\|_p \le C\|\triangle D_t v^n\|_2$;

(3) $\|D_t v_{i,j}^n v_{j,ik}^n\|_p \le C\|\triangle D_t v^n\|_2^{3/p-2}$;

(4) $\|v_{i,j}^n D_t v_{j,i}^n\|_p \le C$;

*where constants C depend on $\nu, \varphi, T, p$ only.*

**Proof.** Apply Hölder's inequality. Then,

$$\int |h_{i,j}g_{j,ik}|^p dx \leq \left( \int |h_{i,j}|^{2p/(2-p)} dx \right)^{1-p/2} \left( \int |g_{j,ik}|^2 dx \right)^{p/2}. \tag{23}$$

Denote $h = v^n$, $g = v^n$. An exponent $2p/(2-p) \in [2,6]$. Then, the first factor in (23) is estimated by Lemma 1 with an assumption $r = 2$, $s = 6$. Uniform estimate (1) follows from Lemma 6 and Lemma 8. In the same way taking a pair $h = v^n$, $g = D_t v^n$ we get estimate (2). Now, denote $h = D_t v^n$, $g = v^n$. To the first factor from the right-hand side of (23) we apply Lemma A5 relying on $r = 2$, $s = 6$, $t = 3 - 3/p$. The norm $\|\nabla D_t v^n\|_2$ has a uniform estimate with respect to $t$ and $n$ by Lemma 7. Apply the both estimates of this lemma and obtain estimate (3). The other estimates (4) and (1) we prove in the same way. □

*2.5. Successive Approximations of Solutions and Its Estimates: Pressure*

Let $v^n$ be an approximation from formulae (14)–(16). Fix $T$, $T < T_0$ where $T_0$ is the constant from Lemma 1. Consider a hydrodynamical potential

$$P^n(t,x) = \frac{1}{4\pi} \int_{R^3} \frac{v_{i,j}^n(t,y)v_{j,i}^n(t,y)dy}{|x-y|}. \tag{24}$$

A product $v_{i,j}^n v_{j,i}^n \in L_1(R^3) \cap L_3(R^3)$. This follows from estimates of Lemma 6, Lemma 8 and Hölder's inequality. By Lemma A4 on every segment $[0, T]$, we have:

$$\|v_{i,j}^n v_{j,i}^n\|_p \leq C(\nu, \varphi, T, p) = C, \ 1 \leq p \leq 3. \tag{25}$$

Lemma A1 implies a uniform estimate with respect to $t$ and $n$:

$$\|P^n\|_q \leq A(p,q)C(\nu, \varphi, T, p) \tag{26}$$

for any exponent $q > 3$, where $\frac{1}{q} = \frac{1}{p} - \frac{2}{3}$.

Let us decompose integral in (24) by two integrals $J_1$ and $J_2$: over ball $|y - x| < 1$ and over its exterior. Every integral we estimate by Hölder's inequality or a simple estimation. Then,

$$4\pi J_1 \leq \|v_{i,j}^n v_{j,i}^n\|_3 \left( \int_{|y-x|<1} \frac{dy}{|x-y|^{1,5}} \right)^{2/3} \leq C_1, \ 4\pi J_2 \leq \|v_{i,j}^n v_{j,i}^n\|_1 \leq C_2.$$

Thus, with some constant $C = C(\nu, \varphi, T)$ on the set $S_T$ for all $n$, we obtain:

$$|P^n(t,x)| \leq C. \tag{27}$$

Function $P^n$ has derivatives in distributions:

$$P_{,i}^n, \ D_t P^n, \ D_t P_{,i}^n, \ P_{,ij}^n, \ D_t P_{,ij}^n.$$

The differentiation of the integral from (24), the summation and a simple estimation give:

$$|\nabla P^n(t,x)| \leq \frac{1}{2\pi} \int \frac{|\nabla v^n(t,y)|^2 dy}{|x-y|^2}, \ |\nabla D_t P^n(t,x)| \leq \frac{1}{2\pi} \int \frac{|v_{i,j}^n(t,y)||D_t v_{j,i}^n(t,y)|dy}{|x-y|^2}. \tag{28}$$

By Lemma A1 for exponents $p \in (1,3]$ and $q > 3/2$ where $\frac{1}{q} = \frac{1}{p} - \frac{1}{3}$, we have:

$$\|\nabla P^n(t,x)\|_q \leq 2A(p,q)\||\nabla v^n|^2\|_p. \tag{29}$$

The right-hand side of (29) is bounded upper by a constant $C = C(\nu, \varphi, T, p)$. Here, we apply inequalities from Lemma 6, Lemma 8 and Lemma A5. Therefore,

$$\|\nabla P^n(t, x)\|_q \leq C. \tag{30}$$

Derivatives

$$v^n_{i,j}, \; D_t v^n_{i,j} \in L_6(R^3) \bigcap L_2(R^3).$$

Thus, $D_t \nabla P^n \in L_q(R^3)$ for any exponent $q > 3/2$. By Lemma A1, we obtain:

$$\|\nabla D_t P^n\|_q \leq 2A(p,q)\|v^n_{i,j} D_t v^n_{j,i}\|_p. \tag{31}$$

Consider two cases: $1 < p \leq 3/2$ and $3/2 < p \leq 3$.

Let $1 < p \leq 3/2$. Then, the right-hand side of (31) is bounded by a constant $C = C(\nu, \varphi, T, p)$. This follows from estimate 4 of Lemma 12.

Let $3/2 < p \leq 3$. Then, the exponent $6p/(6-p) \in [2,6]$. Applying Hölder's inequality, we get

$$\|v^n_{i,j} D_t v^n_{j,i}\|_p \leq \|v^n_{i,j}\|_{6p/(6-p)} \|D_t v^n_{j,i}\|_6.$$

The first factor is estimated uniformly by a some constant $C = C(\nu, \varphi, T, p)$. This is proved by application Lemma A5, Lemmas 6 and 8. The second factor is estimated by inequality (2) from Lemma 7. Hence, for an exponent $q$, $q > 3$, $\frac{1}{q} = \frac{1}{p} - \frac{1}{3}$, we get:

$$\|v^n_{i,j} D_t v^n_{j,i}\|_p \leq C\|\triangle D_t v^n\|_2.$$

Applying the integral representation for derivative $D_t P^n$ in the same way we prove another uniform estimate $\|D_t P^n\|_q \leq C$ for every exponent $q$, $q > 3$.

As the final result from (26), (27), (30), (31), we obtain the following statement.

**Lemma 13.** *Let $T_0$ be a constant from Lemma 1. Let $P^n$ be a function defined by (24). Then, on every segment $[0,T]$, $T < T_0$, with some constants $C_1 = C(\nu, \varphi, T)$, $C_2 = C(\nu, \varphi, T, q)$, there are fulfilled uniform estimates with respect to $t \in [0,T]$ and n:*

(1)  $|P^n(t,x)| \leq C_1$ *for all $x \in R^3$;*
(2)  $\|\nabla P^n\|_q \leq C_2$ *for every $q > 3/2$;*
(3)  $\|\nabla D_t P^n\|_q \leq C_2$ *for every $q \in (3/2, 3]$;*
(4)  $\|\nabla D_t P^n\|_q \leq C_2\|\triangle D_t P^n\|_2$, $\|P^n\|_q \leq C_2$, $\|D_t P^n\|_q \leq C_2$ *for every $q > 3$.*

**Lemma 14.** *Suppose that $T_0$ is the constant from Lemma 1. Let $P^n$ be a function defined by (24). Then, on every segment $[0,T]$, $T < T_0$, with some constants $C_2 = C(\nu, \varphi, T, q)$, there are fulfilled uniform estimates with respect to $t \in [0,T]$ and n:*

(1)  $\|P^n_{,km}\|_q \leq C_2$;
(2)  $\|D_t P^n_{,km}\|_q \leq C_2 \max(1, \|\triangle D_t v^n\|_2)$;

*for every $q \in (3/2, 3]$ and every pair of numbers $k, m = 1, 2, 3$.*

**Proof.** These estimates follow from Lemma A1, Lemma 12 and integral representations for derivatives extracting from (24). Apply Lemma A1 and item (1) of Lemma 12. Then, we obtain the first inequality. In the same way, we get the second inequality with an application of estimates (2) and (3) from Lemma 12. □

**Lemma 15.** *Let $T_0$ be a constant of Lemma 1. Let $P^n$ be a function defined by (24). Then, on every segment $[0, T]$, $T < T_0$, with some constant $C = C(\nu, \varphi, T, q)$, there are fulfilled uniform estimates with respect to $t \in [0, T]$ and n:* $\|P^n_{,klm}\|_q \leq C$ *for every $q \in (1, 3/2]$, $k, l, m = 1, 2, 3$.*

**Proof.** It is sufficient to repeat the proof of Lemma 11 with the application of formula (24).   □

**Lemma 16.** *Let $T_0$ be a constant from Lemma 1. Supposing that $P^n$ is the function defined by (24), then*

$$\triangle P^n = -v^n_{i,j} v^n_{j,i}.$$

**Proof.** This follows from proposition A3.   □

*2.6. Estimates of Uniform Continuity of Approximations in Spaces $L_2(R^3)$ and $C(S_T)$*

Now, we estimate the integral continuity modulus of gradients and Laplacians for approximations following [7]. Let $T_0$ be a constant from Lemma 1. Let $T$, $T_1$ be arbitrary numbers such that $T < T_1 < T_0$. Assume $t \in [0, T]$, $t + h \in [0, T_1]$. Equations (16) we write by the following form:

$$\left( D_t v^n(t + h, \cdot), a^q \right) - \nu \left( \triangle v^n(t + h, \cdot), a^q \right) + \int v^n_i(t + h, x) v^n_{k,i}(t + h, x) a^q_k dx = 0, \qquad (32)$$

$$q = 1, \ldots, n.$$

Every equality we multiply by difference $c_{qn}(t + h) - c_{qn}(t)$ respectively and add together them. Setting $z = v^n(t + h, x) - v^n(t, x)$, we have

$$\left( \frac{\partial z}{\partial h}, \triangle z \right) = \nu(\triangle v^n(t + h, \cdot), \triangle z) - \int v^n_i(t + h, x) v^n_{k,i}(t + h, x) \triangle z_k dx.$$

To the scalar product on the right-hand side, we apply Cauchy–Bunyakovskii's inequality. The integral ($J$ is its mean) we estimate by Corollary A4 for the triple $v^n$, $v^n$, $z$. Then,

$$|(\triangle v^n(t + h, \cdot), \triangle z)| \leq \|\triangle v^n\|_2 \|\triangle z\|_2,$$

$$|J| \leq a\|\nabla v^n\|_2^{3/2} \|\triangle v^n\|_2^{1/2} \|\triangle z\|_2.$$

Every factor from the right-hand side of these inequalities is bounded by a constant $C = C(\nu, \varphi, T_1)$ uniformly with respect to $t$, $n$, $h$. This follows from estimates of Lemmas 6–8, definition of $z$ and the choice of means $h$, $T_1$. Since

$$\left( \frac{\partial z}{\partial h}, \triangle z \right) = -\frac{1}{2} \frac{d}{dh} \|\nabla z\|_2^2,$$

then we get inequalities:

$$-C \leq \frac{1}{2} \frac{d}{dh} \|\nabla z\|_2^2 \leq C.$$

Integrating it over segments $[0, h]$ if $h > 0$ and $[h, 0]$ if $h < 0$ in any case we have: $\|\nabla z\|_2^2 \leq 2C|h|$. Thus, the following statement is proved.

**Lemma 17.** *Let $T_0$ be a constant from Lemma 1. Let $T$, $T_1$, $T < T_1 < T_0$ be arbitrary but fixed numbers. Then, there exists a constant $C = C(\nu, \varphi, T_1)$ such that, for all approximations $v^n$, there is a fulfilled inequality:*

$$\|\nabla v^n(t + h, \cdot) - \nabla v^n(t, \cdot)\|_2 \leq C\sqrt{|h|},$$

*whenever $t \in [0, T]$, $t + h \in [0, T_1]$.*

**Lemma 18.** *Let $T_0$ be a constant from 1. Let $T, T_1, \ T < T_1 < T_0$ be arbitrary but fixed numbers. Then, there exists a constant $C = C(\nu, \varphi, T_1)$ such that for all approximations $v^n$ there is fulfilled inequality:*

$$\|\triangle v^n(t+h, \cdot) - \triangle v^n(t, \cdot)\|_2 \le C \sqrt[8]{|h|^3}$$

*whenever $t \in [0, T]$, $t + h \in [0, T_1]$.*

**Proof.** Formulae (16) and (32) yield equalities:

$$\left(D_t z, a^q\right) - \nu\left(\triangle z, a^q\right) + \int (z_i v^n_{k,i}(t+h, x) + v^n_i(t, x) z_{k,i}) a^q_k dx = 0, \ q = 1, \dots, n,$$

where $z = v^n(t+h, x) - v^n(t, x)$. Every equality we multiply by factor $c\prime_{qn}(t+h)$ respectively and add together them. Furthermore, in the second term, we replace differentiation on variable $t$ by differentiation on variable $h$. Hence, we obtain:

$$\left(D_t z, \triangle D_t v^n(t+h, \cdot)\right) - \nu\left(\triangle z, \frac{\partial}{\partial h}\triangle z\right) =$$

$$- \int (z_i v^n_{k,i}(t+h, x) + v^n_i(t, x) z_{k,i})\triangle D_t v^n_k(t+h, x) dx = -L_1 - L_2.$$

Here, $L_1, L_2$ are integrals from the first and the second products sums, respectively. Hence,

$$\left(\nabla D_t z, \nabla D_t v^n(t+h, \cdot)\right) + \frac{\nu}{2}\frac{\partial}{\partial h}\|\triangle z\|_2^2 = L_1 + L_2. \tag{33}$$

The scalar products on the left-hand side of (33) are bounded uniformly. This follows from estimates of Lemmas 6–8. Therefore, we have:

$$-C - L_1 - L_2 \le \frac{\nu}{2}\frac{\partial}{\partial h}\|\triangle z\|_2^2 = C + L_1 + L_2 \tag{34}$$

with some constant $C = C(\nu, \varphi, T_1)$. A uniform boundedness of integrals $L_1, L_2$ follows from Corollary A4. For the verification, we take mappings triples $z$, $v^n(t+h, \cdot)$, $D_t v^n(t+h, \cdot)$ and $v^n(t, \cdot)$, $z$, $D_t v^n(t+h, \cdot)$, respectively. Finally, applying estimates from Lemma 6, Lemma 8 and Lemma 17, we obtain:

$$|L_1| \le a\|\nabla z\|_2\|\nabla v^n(t+h, \cdot)\|_2^{1/2}\|\triangle D_t v^n(t+h, \cdot)\|_2^{1/2}\|\triangle z\|_2 \le \tag{35}$$

$$C_1\sqrt{|h|}\|\triangle D_t v^n(t+h, \cdot)\|_2^{1/2},$$

$$|L_2| \le a\|\nabla v^n\|_2\|\nabla z\|_2^{1/2}\|\triangle z\|_2^{1/2}\|\triangle D_t v^n(t+h, \cdot)\|_2 \le C_2\sqrt[4]{|h|}\|\triangle D_t v^n(t+h, \cdot)\|_2, \tag{36}$$

where constants $C_m = C_m(\nu, \varphi, T_1)$, $m = 1, 2$ depend on $\nu, \varphi, T_1$ only.
We integrate (34) over segments $[0, h]$ if $h > 0$ and $[h, 0]$ if $h < 0$. Assume $h > 0$ without restriction of the generality. Then, from (35) after Hölder's inequality application and inequality (2) of Lemma 8, we get:

$$\int_0^h |L_1| dh \le C_1 h^{5/4}\left(\int_0^h \|\triangle D_t v^n(t+h, \cdot)\|_2^2 dh\right)^{1/4} \le C_1\sqrt[4]{l_1} h^{5/4},$$

where $C_1$ is a new constant. From (36) in the same way, we obtain another estimate:

$$\int_0^h |L_2| dh \le C_2\sqrt{l_1} h^{3/4}.$$

Integrating (34) and, gathering last estimates, we get lemma inequality. □

**Lemma 19.** *Let $T_0$ be a number from Lemma 1. Let $T$, $T_1$, $T < T_1 < T_0$ be arbitrary but fixed numbers. Then, there exists a constant $C = C(\nu, \varphi, T_1)$ such that for all approximations $v^n$ and $P^n$ there are fulfilled inequalities:*

$$|v^n(t+h,x) - v^n(t,z)| \leq C(|h|^{0,375} + |x-z|^{0,5}),$$

$$|P^n(t+h,x) - P^n(t,z)| \leq C(|h|^{0,375} + |x-z|^{0,5})$$

*whenever $t \in [0,T]$, $t + h \in [0, T_1]$, $|h| \leq 1$, $x, z \in R^3$.*

**Proof.** We have $|f(t+h,x) - f(t,z)| \leq |f(t+h,x) - f(t,x)| + |f(t,x) - f(t,z)|$. Therefore, one should find uniform estimates for every modulus on the right-hand side considering mappings $v^n$, $P^n$. From representation (A2), it follows: $|v^n(t+h,x) - v^n(t,x)| \leq \frac{1}{4\pi}(J_1 + J_2)$, where

$$J_1 = \int_{|y-x|\leq 1} \frac{|\nabla v^n(t+h,\,y) - \nabla v^n(t,\,y)|dy}{|x-y|^2},$$

$$J_2 = \int_{|y-x|\geq 1} \frac{|\nabla v^n(t+h,\,y) - \nabla v^n(t,\,y)|dy}{|x-y|^2}.$$

To every integral, we apply again Hölder's inequality. Then,

$$J_1 \leq \|\nabla v^n(t+h,\cdot) - \nabla v^n(t,\cdot)\|_6 \left(\int_{|y-x|\leq 1} |x-y|^{-12/5}dy\right)^{5/6},$$

$$J_2 \leq \|\nabla v^n(t+h,\cdot) - \nabla v^n(t,\cdot)\|_2 \left(\int_{|y-x|\geq 1} |x-y|^{-4}dy\right)^{1/2}.$$

The second representation in (A2) and Lemma A1 yield estimate:

$$\|\nabla v^n(t+h,\cdot) - \nabla v^n(t,\cdot)\|_6 \leq A\|\triangle v^n(t+h,\cdot) - \triangle v^n(t,\cdot)\|_2.$$

Therefore, previous inequalities and estimates from Lemma 17 and Lemma 18 give formula:

$$|v^n(t+h,x) - v^n(t,x)| \leq C|h|^{0,375}, \tag{37}$$

where $C$ is a constant depending on $\nu$, $\varphi, T_1$ only.

Let us estimate the second modulus applying Poisson's formula (see (A1)). Then,

$$|v^n(t,x) - v^n(t,z)| \leq \frac{|x-z|}{4\pi} \int \frac{|\triangle v^n(t,y)|dy}{|x-y||z-y|} = \frac{|x-z|}{4\pi} J_3.$$

From the inequality,

$$J_3 \leq \|\triangle v^n(t,\cdot)\|_2 \left(\int |x-y|^{-2}|z-y|^{-2}dy\right)^{1/2},$$

with some constant $C_1$, we obtain:

$$J_3 \leq C_1 \|\triangle v^n(t,\cdot)\|_2 |x-z|^{-1/2}.$$

Previous estimates and Lemma 8 (estimate (1)) yield:

$$|v^n(t,x) - v^n(t,z)| \leq C|x-z|^{0,5}, \tag{38}$$

where a constant $C$ depends on $\nu, \varphi, T_1$ only. Thus, the first estimate follows from (37) and (38).

In the same way, we prove an inequality of the kind (38) for the function $P^n$ (formula (24)). The norm $\|\nabla v^n\|_4$, which appears after applying Holder's inequality, we must estimate by Lemma A5. Then,

$$\|\nabla v^n\|_4 \le \|\nabla v^n\|_2^{1/2}\|\nabla v^n\|_6^{1/6}.$$

Furthermore, Lemma 6 (estimates (1), (2)) and lemma 8 (estimate (1)) yield the inequality $\|\nabla v^n\|_4 \le C$, where $C = C(\nu, \varphi, T)$ is some universal constant. Then, it follows:

$$|P^n(t,x) - P^n(t,z)| \le C_1|x-z|^{0,5}. \tag{39}$$

A difference $L = P^n(t+h,x) - P^n(t,x)$ is represented in the following form:

$$L = \frac{1}{4\pi} \int \frac{(v_{i,j}^n(t+h,y) - v_{i,j}^n(t,y))(v_{j,i}^n(t+h,y) - v_{j,i}^n(t,y))dy}{|x-y|}.$$

To obtain this formula, we change summation index for a separate terms (use (24)) and apply Hölder's inequality for three factors and two factors. We make estimates separately on a ball $|y-x| \le 1$ and its exterior. Let $m = \left(\int_{|y-x|\le 1} |x-y|^{-2}dy\right)^{1/2}$. Then,

$$\left|\int_{|y-x|\le 1} (\cdot)dy\right| \le m\|\nabla v^n(t+h,\cdot) - \nabla v^n(t,\cdot)\|_6(\|\nabla v^n(t+h,\cdot)\|_3 + \|\nabla v^n(t,\cdot)\|_3),$$

$$\left|\int_{|y-x|\ge 1} (\cdot)dy\right| \le \|\nabla v^n(t+h,\cdot) - \nabla v^n(t,\cdot)\|_2(\|\nabla v^n(t+h,\cdot)\|_2 + \|\nabla v^n(t,\cdot)\|_2).$$

In the last case, as the first step, we make a simple estimate, thereupon, we apply Hölder's inequality. The analogous arguments that are used above for the proof of the first estimate in lemma and formula (39) yield the inequality:

$$|P^n(t+h,x) - P^n(t,x)| \le C|h|^{0,375}, \tag{40}$$

where $C = C(\nu, \varphi, T_1)$ is some constant depending on $\nu, \varphi, T_1$ only. Uniform estimates (39) and (40) prove the second inequality of lemma. $\square$

*2.7. Weak Limits Properties of Approximation Sequences*

**Lemma 20.** *Let $T_0$ be a number from Lemma 1 and $T < T_0$ be a positive number. Then, the sequence of mappings $(v^n)_{n=1,\dots}$ defined by (14)–(16) is bounded in the space $W_6^1(S_T)$ and the sequence $(P^n)_{n=1,\dots}$ constructed by formula (24) is bounded in spaces $W_q^1(S_T)$, $q > 3$.*

**Proof.** Estimate (2) from Lemma 6 and estimate (1) from Lemma 8 yield inequality $\|\nabla v^n\|_6 \le C$. It is fulfilled with some constant $C$ whenever $n$ and $t \in [0,T]$. For all mappings $v^n$, $D_t v^n$ integral representation (A2) is true. Then, by Lemma A1, we obtain:

$$\|v^n\|_6 \le A\|\nabla v^n\|_2, \ \|D_t v^n\|_6 \le A\|\nabla D_t v^n\|_2.$$

From inequalities (1) of Lemmas 6 and 8, we conclude that there exist constants $C_1, C_2$, such that $\|v^n\|_6 \le C_1$, $\|D_t v^n\|_6 \le C_2$. All norms are uniformly bounded with respect to $t$. Hence, the sequence $(v^n)_{n=1,\dots}$ is bounded in $W_6^1(S_T)$.

Uniform boundedness of these norms $\|P^n\|_q$, $\|\nabla P^n\|_q$, $\|D_t P^n\|_q$, $q > 3$, with respect to $t$ and $n$ follows from Lemma 13. Therefore, the sequence $(P^n)_{n=1,\dots}$ is bounded in spaces $W_q^1(S_T)$. $\square$

**Remark 2.** *The spaces $W_6^1(S_T)$, $W_q^1(S_T)$ are reflexive. Hence, every bounded set from it is a weakly compact set (see [? ]). Then, by Lemma 20, sequences $(v^n)_{\ldots}$, $(P^n)_{\ldots}$ are bounded in these spaces. It is possible to extract a weakly converging subsequences from its. Let*

$$u(t,x) = \lim_{k \to \infty} v^{n_k}(t,x), \; P(t,x) = \lim_{k \to \infty} P^{n_k}(t,x) \tag{41}$$

*be weak limits of these subsequences. Without restriction of generality, we assume that these subsequences converge to the own weak limits on every compact set of $S_T$. This follows from Arzela's theorem and Lemma 19.*

**Lemma 21.** *Let u and P be weak limits from (41). Then,*

*(1)    mappings u and P are uniformly continuous on a set $S_T$, $T < T_0$, moreover, $u(0,x) = \varphi(x)$;*

*(2)    mappings u and P are bounded on a set $S_T$;*

*(3)    the mapping $u \in W_6^1(S_T)$ and there exists a constant $C = C(\nu, \varphi, T)$ such that following inequalities are true: $\|u\|_6 \leq C$, $\|\nabla u\|_6 \leq C$, $\|D_t u\|_6 \leq C$ whenever $t \in [0,T]$;*

*(4)    $\|\nabla u\|_2 \leq C\|\nabla \varphi\|_2$, $\|\nabla D_t u\|_2 \leq C\|\nabla T^1\|_2$ whenever $t \in [0,T]$, where vector field $T^1$ from (22), a constant $C = C(\nu, \varphi, T)$;*

*(5)    u has distributions of the second and third orders: $u_{,ij}$, $D_t u_{,ij}$, in addition, for all $t \in [0,T]$, there are fulfilled inequalities: $\|\triangle u\|_2 \leq C$, $\int_0^t \|\triangle D^t u\|_2^2 dt \leq l_1$ where constants $C, l_1$ from Lemma 8;*

*(6)    the function $P \in W_q^1(S_T)$ for every $q > 3$, in this case, there exists a constant $C = C(\nu, \varphi, T, q)$ such that, for all $t \in [0,T]$ estimates $\|P\|_q \leq C$, $\|D_t P\|_q \leq C$ are true;*

*(7)    there exist constants $C_i = C_i(\nu, \varphi, T, q)$ such that $\|\nabla P\|_q \leq C_1$ for every $q > 3/2$ and $\|\nabla D_t P\|_q \leq C_2$ for every $q \in (3/2, 3]$;*

*(8)    the function P has distributions of the second and third orders: $P_{,km}$, $P_{,kmj}$, $D_t P_{,i}$, in addition, there exists a number $C = C(\nu, \varphi, T, q)$ such that, for all $t \in [0,T]$, the following inequalities hold:*

$$\|P_{,km}\|_q \leq C, \; \|D_t P_{,i}\|_q \leq C \text{ for every } q \in (3/2, 3] \text{ and } \|P_{,kmj}\|_q \leq C, \text{ for every } q \in (1, 3/2].$$

**Proof.** Property (1) follows from Remark 2. A uniform continuity follows from Lemma 19 and a uniform convergence of subsequences $(v^{n_k})_{k=1,\ldots}$ and $(P^{n_k})_{k=1,\ldots}$ on compact subsets of $S_T$.

Property (2) follows from a uniform convergence on compact sets, Lemma 9 and Lemma 13 (item (1)).

Property (3) follows from norm semicontinuity of a weak limit in reflexive spaces.

Property (4) follows from Lemma 11. A uniform boundedness of norms $\|v^n_{,ij}\|_2$ (see Lemma 8 and Lemma 11) and norms boundedness $\|D_t v^n_{,ij}\|_2$ in the space $W_2^1(S_T)$ (see Lemma 8 and Lemma 11) guarantee an existence of distributions $u_{,ij}$, $D_t u_{,ij}$. Estimates of its norms follow from a semicontinuity of a weak limit norm.

Properties (5)–(8) are proved in the same way. For the verification, we apply Lemmas 13–15.    □

**Lemma 22.** *Weak limits from (41) satisfy equalities:*

$$P_{,k} = -\frac{1}{4\pi} \int \frac{u_{i,j}(t,y)u_{j,i}(t,y)(x_k - y_k)dy}{|x-y|^3}, \; u_{,j} = \frac{1}{4\pi} \int \frac{\triangle u(t,y)(x_j - y_j)dy}{|x-y|^3}.$$

**Proof.** The first equality is fulfilled for mappings $v^n$ and $P^n$. The sequence $(\nabla v^n)_{n=1,\ldots}$ is bounded in the space $W_2^1(S_T)$. In addition, estimates of norms $\|\nabla v^n\|_2$, $\|v^n_{,ij}\|_2$, $\|D_t v^n_{,i}\|_2$ are uniform with respect to $t$ and $n$ (see Lemmas 6–8 and 11). Apply Sobolev–Kondrashov's embedding theorem (see [23], pp. 83–94) to the sequence $(\nabla v^n)_{n=1,\ldots}$. As a bounded set, it is embedded in the space $L_q([0,T] \times \Omega)$ for every ball $\Omega \subset R^3$. An exponent $q$ satisfies condition

$$\frac{1}{q} - \frac{1}{2} + \frac{1}{m} > 0, \; q < 4.$$

In this case, a dimension of spatial domain $[0, T] \times \Omega)$ $m = 4$. Thus, we can assume that a subsequence $(\nabla v^{n_k})_{k=1,...}$ converges strongly to a mapping $\nabla u$ in the space $L_q([0, T] \times \Omega)$, $q < 4$, for every ball $\Omega \subset R^3$. Denote the integral from the first equality of the lemma by $Q_k(t, x)$. Let $d^n = P^n_{,m} - Q_m$. From equality

$$v^n_{i,j} v^n_{j,i} - u_{i,j} u_{j,i} = (v^n_{i,j} - u_{i,j})(v^n_{j,i} + u_{j,i}),$$

we deduce:

$$|d^n(t, x)| \leq \frac{1}{4\pi} \int \frac{|\nabla v^n(t, y) - \nabla u(t, y)||\nabla v^n(t, y) + \nabla u(t, y)| dy}{|x - y|^2}.$$

Multiply this inequality by $|\eta|$ where $\eta \in C_0(S_T)$ an arbitrary test–function. Thereupon, integrate over the set $S_T$ and change integration order. Then,

$$\left| \int_{S_T} d^n \eta dx \right| \leq \int_0^T \int_{R^3} I_2(|\eta|) |\nabla v^n - \nabla u| |\nabla v^n + \nabla u| dy dt = \int_0^T (K_1 + K_2) dt,$$

where $I_2$ is the Riesz potential, $K_1$ is the interior integral calculating over ball $|y| < r$, and $K_2$ is the interior integral calculating over exterior of this ball. Estimate every integral applying Hölder's inequality. Thus, we have

$$K_2 \leq \left( \int_{|y| \geq r} I_2(|\eta|) dy \right)^{1/2} \|\nabla v^n - \nabla u\|_3 \|\nabla v^n + \nabla u\|_6.$$

The second and the third factors on the right-hand side we estimate by constants independent of $t$ and $n$ (see Lemmas 6, 8, 21 with conditions (3)–(4) and Lemma A5). A radius $r$ is fixed so that the first factor is less an arbitrary positive number $\varepsilon$. Then, $K_2 \leq C\varepsilon$. Integral $K_1$ we estimate on a subsequence. Then,

$$K_1 \leq \left( \int_{|y| \leq r} |\nabla v^{n_k} - \nabla u|^3 dy \right)^{1/3} \|I_2(|\eta|)\|_2 \|\nabla v^n + \nabla u\|_6.$$

The second and the third factors are uniformly bounded by a some constant $C$. Therefore, the inequality:

$$\int_0^T K_1 dt \leq C \left( \int_0^T \int_{|y| < r} |\nabla v^{n_k} - \nabla u|^3 dy dt \right)^{1/3} \sqrt[3]{T^2}$$

is fulfilled. The middle factor is not greater $\varepsilon$ if a number $k$ is large enough. This follows from condition of a strong convergence on a bounded set. Combining all estimates above, we obtain the inequality

$$\left| \int_{S_T} d^{n_k} \eta dx \right| \leq C\varepsilon T + C\varepsilon \sqrt[3]{T^2}.$$

This means that $d^{n_k} \to 0$ weakly because a function $\eta$ is an arbitrary. The first equality is proved. The second equality is proved in the same way. Consider the difference $d^n = v^n_{,j} - R_j$ where $R_j$ is the integral of the second equality. In the integral $d^n(t, x)$, we replace the variable by $y = x + z$. Thereupon, we multiply the equality by a test–function $\eta \in C_0(S_T)$ and integrate its over set $S_T$. Change integration order and carry over Laplace operator to function $\eta$. Then,

$$\left| \int_{S_T} d^n \eta dx dt \right| \leq \frac{1}{4\pi} \int_0^T \int_{R^3} \frac{1}{|z|^2} \int_{R^3} |v^n(t, x + z) - u(t, x + z)| |\triangle \eta(t, x)| dx dz dt.$$

Replace variables in the interior integral by $x = y - z$ and change integration order. Hence, we get:

$$\left| \int_{S_T} d^n \eta dx dt \right| \leq \int_0^T \int_{R^3} |v^n(t, y) - u(t, y)| |I_1(\triangle \eta)(y)| dy dt.$$

Integration with respect to $y$ we make separately over ball $|y| < r$ and its exterior. The furthest arguments are conducted in the same way as above. A distinction in the following. In this time, we use an uniform convergence of a subsequence $(v^{n_k})_{k=1,\dots}$ on compact sets (see Remark 2). □

*2.8. Weak Solutions and Gradients Boundedness*

**Lemma 23.** *Let $u$ and $P$ be weak limits from* (41). *Then, for every solenoidal vector field $\psi \in C_0^\infty(R^3)$ and almost everywhere $t \in [0, T]$, there is fulfilled integral identity:*

$$(D_t u, \triangle \psi) - \nu(\triangle u, \triangle \psi) + \int u_i u_{j,\,i} \triangle \psi_j dx + (\nabla P, \triangle \psi) = 0.$$

**Proof.** Equalities (16) multiply by a test–function $\eta \in C_0^\infty([0, T])$ and integrate its over segment $[0, T]$. If a subsequence $(v^{n_k})_{k=1,\dots}$ converges weakly, then, for all $q = 1, \dots, n_k$, we have

$$\int_0^T (D_t v^{n_k}, \triangle a^q)\eta(t)dt - \nu \int_0^T (\triangle v^{n_k}, \triangle a^q)\eta(t)dt + \int_0^T \int v_i^{n_k} v_{j,\,i}^{n_k} \triangle a_j^q \eta(t)dxdt = 0.$$

Fix a some number $q$. Then, the passage to the limit gives the equality

$$\int_0^T (D_t u, \triangle a^q)\eta(t)dt - \nu \int_0^T (\triangle u, \triangle a^q)\eta(t)dt + \int_0^T \int u_i u_{j,\,i} \triangle a_j^q \eta(t)dxdt = 0. \qquad (42)$$

This is explained by a weak convergence of a sequence $(v_i^{n_k} v_{,\,i}^{n_k})_{k=1,\dots}$ to the mapping $u_i u_{,\,i}$. It is given by support compactness of a vector field $a^q$, by uniform boundedness with respect to $t$ and $n$ of norms $\|\nabla v^n\|_p$, $2 \le p \le 6$, and a uniform convergence of subsequence $(v^{n_k})_{k=1,\dots}$ on compact subsets of $S_T$. A function $\eta$ is an arbitrary. Therefore, from (42), we obtain

$$(D_t u, \triangle a^q) - \nu(\triangle u, \triangle a^q) + \int u_i u_{j,\,i} \triangle a_j^q dx = 0.$$

It is already fulfilled for every natural number $q$. The construction of vector fields $a^q$ permits this integral identity to extend on elements of the fundamental system $(\psi_{n=1,\dots}^n)$ (see (12) and (13)), i.e.,

$$(D_t u, \triangle \psi^n) - \nu(\triangle u, \ \triangle \psi^n) + \int u_i u_{j,\,i} \triangle \psi_j^n dx = 0. \qquad (43)$$

We show that identity (43) is true for every solenoidal vector field $\psi \in C_0^\infty(R^3)$. Let $(\xi^m)_{m=1,\dots}$ be a sequence of a finite linear combinations of mappings $\psi^n$, which converges to a vector field $\psi \in C_0^\infty(R^3)$ in the space $J_0^2(R^3)$. Then,

$$\|\nabla \xi^m - \nabla \psi\|_2 \to 0, \ \|\xi_{,\,ij}^m - \psi_{,\,ij}\|_2 \to 0$$

and equality (43) for mappings $\xi^m$ is true. Mappings $\triangle u, \ u_i u_{,\,i}$ belong to the space $L_2(R^3)$ for a.e. $t$. Then,

$$(\triangle u, \triangle \xi^m) \to \triangle u, \triangle \psi), \int u_i u_{j,\,i} \triangle \xi_j^m dx \to \int u_i u_{j,\,i} \triangle \psi_j dx$$

a.e. as $m \to \infty$. Let us show

$$(D_t u, \triangle \xi^m) \to (D_t u, \triangle \psi)$$

as the same condition is. Consider the equality of scalar products

$$-(D_t u, \triangle \xi^m) = (D_t u_{,\,j}, \xi_{,\,j}^m)$$

and note that the right side tends to $(D_t u_{,\,j}, \psi_{,\,j})$ (see Lemma 21 item (4)). On the other side, $-(D_t u_{,\,j}, \psi_{,\,j}) = (D_t u, \triangle \psi)$. Condition (43) is true for an arbitrary $\psi \in C_0^\infty(R^3)$. From $(\nabla P, \triangle \psi) = 0$, we have the lemma. □

**Lemma 24.** *(see ([7], pp. 41–44), see also [29].) Let $B \subset R^3$ be an arbitrary ball. Then, a space $L_2(B)$ of any vector fields has a decomposition by a direct sum $L_2(B) = G(B) \oplus J_0(B)$ of orthogonal subspaces. A subspace $G(B)$ is the space of gradients $\nabla g$ where $g : B \to R$ is locally square–integrable function with a finite norm $\|\nabla g\|_2$. A space $J_0(B)$ is the closure with respect to the norm $L_2(B)$ of all solenoidal vector fields from the class $C_0^\infty(B)$.*

**Lemma 25.** *If u and P are weak limits (41), then there are fulfilled equalities:*

$$D_t u_k - \nu \triangle u_k + u_i u_{k,\,i} + P_{,\,k} = 0, \ k = 1, 2, 3,$$

*a.e. on a set $S_T$ for any $T \in [0, T_0)$.*

**Proof.** Let

$$H_k = D_t u_k - \nu \triangle u_k + u_i u_{k,\,i} + P_{,\,k}.$$

Denote $h_2 = -\nu \triangle u$, $h_3 = u_i u_{,\,i}$, $h_6 = D_t u + \nabla P$. Every vector field $h_p$, $p = 2, 3, 6$, belongs to the space $L_p(R^3)$ (see Lemma 21). Mappings norms $h_p$ are bounded by constants independent of $t \in [0, T]$. From the first equality of Lemma 22, we gather $(H, \nabla g) = 0$, where $g \in C_0^\infty(R^3)$ is an arbitrary. We assume the mapping $H$ and its generators $h_p$ belong to the class $C^\infty(R^3)$. Otherwise, we take averages with a kernel from $C_0^\infty(R^3)$ for them. For averages, the equality $(H, \nabla g) = 0$ and the equality of Lemma 23 are kept. This follows from behind an arbitrary choice of a smooth function $g$ and a field $\psi \in C_0^\infty(R^3)$. Then, $div\ H = 0$. Moreover, a smoothness $H$ and the equality of Lemma 23 imply $(\triangle H,\ \psi) = 0$. From Lemma 24 on every ball $B \subset R^3$, we have $\triangle H = \nabla h$. A function $h$ is infinitely smooth. This is given by smoothness $\triangle H$. Then, $div\ \triangle H = \triangle h$. On the other hand, $div\ \triangle H = \triangle div\ H = 0$. Therefore, the function $h$ is a harmonic function. Hence, and from above, there is $\triangle^2 H = 0$. By Lemma A7, we have $H = 0$. Making an average parameter tending to zero, we obtain this equality in the general case. $\square$

**Lemma 26.** *Let u and P be weak limits from (41). Then, there exists a number $C = C(\nu, \varphi, T)$ such that, for almost everywhere, $t \in [0, T]$ following conditions are fulfilled:*

(1) $\|\triangle u\|_6 \leq C$;
(2) $|\nabla u_k(t, x)| \leq C$, $|\nabla P(t, x)| \leq C$, $k = 1, 2, 3$.

**Proof.** From Lemma 25, we conclude that Laplacian $\triangle u$ is the linear combination of three vector fields $\nabla P, D_t u, u_i u_{,\,i}$. Coordinates $u_i$ are bounded on the set $S_T$ by Lemma 21 item (2). Then, from Lemma 21 (see estimates (3) and (6)), it follows the first part of the lemma.
Gradients boundness $\nabla u_i$ we obtain from the second integral representation of Lemma 22 and estimate $\|\triangle u\|_6 \leq C$. In the next step, we repeat the proof of Lemma 9.
Gradients boundedness $\nabla P$ we get from the first integral representation of Lemma 22 and gradients boundedness $\nabla u_i$ with repeating of the proof from Lemma 9. $\square$

*2.9. Weak Solutions, Integral Equations and Energetic Inequality*

Let $\Gamma(x, t) = (4\pi \nu t)^{-n/2} e^{-|x|^2/4\nu t}$ be a Weierstrass kernel. Furthermore, we consider mixed norms for mappings defined on the set $S_T = [0, T] \times R^n$.

**Lemma 27.** *(See [13], Theorem 2.1.) A vector field $u : S_T \to R^n$ with a finite mixed norm $\|u\|_{p,\,q}$ is a weak solution of problems (1) and (2) if and only if when u is a solution of integral equation*

$$u + B(u, u) = f, \tag{44}$$

*where B is a some nonlinear integral operator, $f(t, x) = \int \Gamma(x - y, t) \varphi(y) dy$.*

**Lemma 28.** *(See [13], Theorem 3.4.) Let u be a solution of integral Equation (44) with a finite mixed norm* $\|u\|_{p,\,q}$ *where* $p, q \geq 2$, $\frac{3}{p} + \frac{2}{q} \leq 1$. *Let k be a positive integer such that* $k + 1 < p$, $q < \infty$. *If mixed norms of derivatives*

$$D^\alpha \frac{\partial^j f}{\partial t^j}$$

*with exponents* $p_1 = \frac{p}{|\alpha|+2j+1}$, $q_1 = \frac{q}{|\alpha|+2j+1}$ *are finite whenever* $|\alpha| + 2j \leq k$, *then also mixed norms of*

$$D^\alpha \frac{\partial^j u}{\partial t^j}$$

*are finite for the same means* $\alpha, j, p_1, q_1$.

**Remark 3.** *The proof of this result relies on Calderon–Zygmund's theorem and a boundedness of singular integral operators of parabolic type (see [30]).*

**Remark 4.** *Norms* $D^\alpha \frac{\partial^j u}{\partial t^j}$ *are bounded by a constant that depends on exponents p, q, derivative order and the mixed norm* $\|u\|_{p,\,q}$. *It follows directly from the proof of the theorem in [13].*

**Lemma 29.** *If u is a weak limit from (41), then there exists a number* $C = C(\nu, \varphi, T, p, q)$ *such that* $\|u\|_{p,\,q} \leq C$ *whenever p*, $q \geq 2$, $\frac{3}{p} + \frac{2}{q} \leq 1$.

**Proof.** Let $T < T_0$ be a positive arbitrary number. Integrate the equality of Lemma 25 over segment $[0, t]$ where $t < T$. Then, continuity and absolute continuity on lines of mapping $u$ give:

$$u(t, x) - \varphi(x) = \int_0^t (\nu \triangle u(\tau, x) - u_i(\tau, x)u_{,\,i}(\tau, x) - \nabla P(\tau, x))d\tau.$$

Every integrable term has finite norms

$$\|\triangle u\|_2, \ \|\nabla P\|_2, \ \|u_i u_{,\,i}\|_2.$$

In addition, every norm is bounded by a constant $C = C(\nu, \varphi, T)$ depending on $\nu, \varphi, T$ only. It follows from Lemma 21 (see estimates (5) and (7)) for the first and the second norms. A boundedness of the third norm follows from mapping boundedness $u$ (see Lemma 21 item (2)) and the estimate from item (4) (see Lemma 21) . Therefore, $\|u\|_2 \leq C$. A boundedness of vector field $u$ (see Lemma 21 item (2)) gives a uniform estimate $\|u\|_p \leq C$ whenever $p \geq 2$. Then, any mixed norm $\|u\|_{p,\,q}$ is finite whenever $p$, $q$ from lemma condition. $\square$

**Lemma 30.** *If u is a weak limit from (41), then a mixed norm* $\|\triangle u\|_{6/5,\,4} < \infty$.

**Proof.** Let initial data $\varphi \in C^\infty_{6/5,\,3/2}$. Function $f$ from Lemma 27 is represented by integral

$$f(t, x) = \frac{1}{\pi^{3/2}} \int e^{-|z|^2} \varphi(x + \sqrt{4\nu t}z)dz.$$

For $\varphi \in C^\infty_{6/5,\,3/2}$, there is true Lemma 34. Therefore, the mapping $f$ and any of its derivatives have a finite mixed norm $\|\cdot\|_{p,\,q}$. By Lemma 25 and Lemma 29, the vector field $u$ is a weak solution of problems (1) and (2) with a finite mixed norm $\|u\|_{p,\,q}$ whenever $p, q \geq 2$. Then, from Lemma 27, we conclude that $u$ is a solution of integral Equation (44). From Lemma 28, we obtain a finiteness of mixed norms for the second derivatives $\|D^\alpha u\|_{p_1,\,q_1}$, where $p_1 = p/3$, $q_1 = q/3$, $|\alpha| = 2, j = 0$. Let $p = 18/5, q = 12$. Then, we have the statement of the lemma. $\square$

**Lemma 31.** *(Energetic condition.) Let u and P be weak limits from* (41)*. Then,*

$$\|u\|_2^2 + 2\nu \int_0^t \|\nabla u(\tau, x)\|_2^2 d\tau = \|\varphi\|_2^2$$

*for every* $t \in [0, T_0)$ *where* $T_0$ *from Lemma* 1.

**Proof.** Note that weak solutions satisfy conditions:

$$J_1 = \int P_{,k} u_k dx = 0, \quad J_2 = \int u_i u_{k,i} u_k dx = 0. \tag{45}$$

From the first equality of Lemma 22, we have:

$$J_1 = \frac{1}{4\pi} \int_{R^3} u_{i,j}(t,y) u_{j,i}(t,y) \int_{R^3} \frac{u_k(t,x)(x_k - y_k)}{|x-y|^3} dx dy. \tag{46}$$

Integrals commutation is possible since the integral over $R^6$ is a finite. It follows from

$$\int_{R^6} |\cdot| dx dy \le 4\pi \int_{R^3} |\nabla u(\tau, y)|^2 I_1(|u|) dy,$$

Tonnelli's theorem, boundedness and summability of $u$, $\nabla u$ with any exponent not less than two and Lemma A1. Here, $I_1$ is the Riesz potential. The interior integral in (46) is equal to zero since

$$\int_{|x-y|<r} \frac{u_k(t,x)(x_k - y_k)}{|x-y|^3} dx = -\int_{|x-y|=r} \frac{u_k(t,x)(x_k - y_k)}{r^2} dS =$$

$$-\frac{1}{r} \int_{|x-y|<r} div \, u dx = 0$$

for any radius $r$.

Let us prove the second equality from (45). The second equality of Lemma 22 implies:

$$J_2 = \frac{1}{4\pi} \int_{R^3} \triangle u_k(t,y) \int_{R^3} \frac{u_i(t,x) u_k(t,x)(x_i - y_i)}{|x-y|^3} dx dy. \tag{47}$$

Integrals commutation we prove in the same way. There is inequality:

$$\int_{R^6} |\cdot| dx dy \le 4\pi \int_{R^3} |\triangle u(\tau, y)|^2 I_1(|u|^2) dy.$$

The right-hand side is a finite because $\triangle u \in L_p$, $2 \le p \le 6$ (see Lemma 21 item (5), Lemma 26 item (1), Lemma A5). In addition, $I_1(|u|^2) \in L_p$, $p > 3/2$ by Lemma A1. To interior integral in (47) we apply the Stokes formula. Then,

$$\int_{|x-y|<r} \frac{u_i(t,x) u_k(t,x)(x_i - y_i)}{|x-y|^3} dx =$$

$$\int_{|x-y|<r} \frac{u_i(t,x) u_{k,i}(t,x)}{|x-y|} dx - \frac{1}{r} \int_{|x-y|=r} u_i u_k \frac{x_i - y_i}{r} dS.$$

A product $u_i u_k$ belongs to the space $W_p^1(R^3)$ whenever $p > 1$. Then, the integral over surface tends to zero as $r \to \infty$ (to apply Lemma A4 with exponent $\alpha = 1$ and a mean $p$, close to unit). Hence, and from (47) we have:

$$J_2 = \frac{1}{4\pi} \int_{R^3} \triangle u_k(t,y) \int_{R^3} \frac{u_i(t,x) u_{k,i}(t,x)}{|x-y|} dx dy. \tag{48}$$

In the iterated integral

$$\int_{|y|<r} \triangle u_k(t,y) \int_{R^3} \frac{u_i(t,x)u_{k,i}(t,x)}{|x-y|}dxdy,$$

we change integration order because the double integral is finite (see above). Hence, we get:

$$\int_{|y|<r} \triangle u_k(t,y) \int_{R^3} \frac{u_i(t,x)u_{k,i}(t,x)}{|x-y|}dxdy = \tag{49}$$

$$\int_{R^3} u_i(t,x)u_{k,i}(t,x) \int_{|y|<r} \frac{\triangle u_k(t,y)}{|x-y|}dydx.$$

The interior integral in the right-hand side of (49) is uniformly bounded with respect to $r > 1$. This follows from a boundedness of the Riesz potential $I_2(|\triangle u|)$. It is proved in the same way as Lemma 9 with applications Lemma 26 item (1) and Lemma 21 item (5). Furthermore, we use Lebesgue's theorem. Then, (48) and (49) give the equality of iterated integrals:

$$J_2 = \int_{R^3} u_i(t,x)u_{k,i}(t,x)I_2(\triangle u_k)(x)dx. \tag{50}$$

The mapping $u \in J_0^2(R^3)$ (norm defined by (15)). Lemma A1 shows that Poisson's formula is true for elements of the space $J_0^2(R^3)$. Then, $I_2(\triangle u) = -u$. Therefore, we have $J_2 = -J_2$ from (50). The second equality from (45) is proved.

Let us show that vector field $u$ satisfies the equality

$$\int_{R^3} u_k \triangle u_k dx = -\|\nabla u\|_2^2 \tag{51}$$

a.e. on $[0,T]$. We have the equality of iterated integrals:

$$\int_{R^3} \triangle u_k(t,x) \int_{|y|<r} \frac{u_{k,j}(t,x)(x_j - y_j)}{|x-y|^3}dydx = \tag{52}$$

$$\int_{|y|<r} u_{k,j}(t,y) \int_{R^3} \frac{\triangle u_k(t,x)(x_j - y_j)}{|x-y|^3}dydx.$$

A finiteness of double integral follows from a boundedness $\nabla u$ (see Lemma 26, item (2)) and properties of the Riesz potential $I_1(|\triangle u|)$. Let $r \to \infty$. The interior integral on the left-hand side of (52) tends to $4\pi u_k(t,x)$ in the space $L_6(R^3)$ for almost every $t$. (See Lemma A1 and equality (A2), which is true for elements of the space $J_0^2(R^3)$ ). The norm $\|\triangle u\|_{6/5}$ is finite a.e. by Lemma 30. In (52), we make the passage to the limit. The interior integral on the right-hand side of (52) is replaced by application of Lemma 22. Then, we get (51). To finish the proof, we are helped with the following steps. Every equality from Lemma 25 we multiply by function $u_k$. Thereupon, we add together them and integrate over space $R^3$. From (45) and (49), we have

$$(D_t u, u) + \nu\|\nabla u\|_2^2 = 0.$$

Hence, we get the required equality.   $\square$

*2.10. Proof of Theorem 1*

Observe that all estimates in proved lemmas above depend on norms $\|\nabla \varphi\|_2$, $\|\nabla T^1\|_2$ (see (22)), $\|\triangle \varphi\|_2$ or $\|\varphi\|_2$ only and don't depend on a diameter of Laplacian support $\triangle \varphi$.

If $\varphi \in C_{6/5,\,3/2}^\infty$, then, by Lemma A4 integrals,

$$\frac{1}{r^2}\int_{|y-x|=r} |\varphi(y|dS, \frac{1}{r}\int_{|y-x|=r} |\nabla \varphi(y|dS$$

tend to zero as $r \to \infty$. Therefore, equalities from Lemma A2 are true for mappings of the class $C^\infty_{6/5, 3/2}$. In addition, we have summability $\varphi$ with any exponent $p > 6/5$ and $\nabla \varphi$ with any exponent $p > 3/2$ (see Lemma 32).

1. Assume that initial data $\varphi \in C^\infty_{6/5, 3/2}$ and its Laplacian support is a compact set. Let $T_0$ be a constant from Lemma 1. Then, item (3) follows from Lemma 26, and items (1) and (4) we get from Lemma 21.

Let us prove estimates of item (2). A uniform boundedness with respect to $t$ of norms

$$\|u\|_6, \ \|\nabla u\|_6, \ \|D_t u\|_6$$

we obtain from Lemma 21 (item (3)). An uniform boundedness of norms

$$\|\nabla u\|_2, \ \|\nabla D_t u\|_2, \ \|\triangle u\|_2, \|\nabla P\|_2$$

follows from Lemma 21 (see items (4), (5), (7)). The estimate of norm $\|u\|_2$ follows from Lemma 31. A uniform boundedness of norms $\|u_{,ij}\|_6$ we get by Lemma 26. A uniform boundedness for norm $\|D_t u\|_2$ is the corollary of Lemma 25 because $D_t u$ is the finite linear combination of terms with uniform bounded norms in the space $L_2(R^3)$. Uniform estimates of norms in spaces $L_p(R^3)$, $2 < p < 6$ we take from Lemma A5. The occurrence of vector field $u$ in spaces $W^1_2(S_T)$ and $W^1_6(S_T)$ we get from the uniform estimates proved above. By Lemma 25 and Lemma 21 (see items (5) and (7)), we obtain $D^2_{tt} u = \nu \triangle D_t u - u_i D_t u_{,i} - (D_t u_i) u_{,i} - D_t \nabla P$. Hence, it follows a finiteness of norm $\|D^2_{tt} u\|_2$ since $u$ and $\nabla u$ are bounded. Therefore, $u \in W^2_2(S_T)$ .

Let us prove item (5) using mixed norms (see [8,26,27]). Weak solutions $u$ and $P$ belong to class $C(S_T)$ (see item 1) of this theorem). In Lemma 28, we put $p = q$ assuming it is very large. Now, we fix an order of derivatives: $m > 1$. Then, by Lemma 27 and Lemma 28, derivative norms $\|D^\alpha D^j_t u\|$, $|\alpha| \le m$ are bounded in the space $L_r(S_T)$ where an exponent $r \ge 6$ is an arbitrary but fixed. A boundedness of weak solution $u$ and its summability in $L_2(S_T)$ imply the belonging $u \in L_r(S_T)$, $r \ge 2$. Exponents means $r$, $p = q$ we choose by large numbers so that the next conditions are fulfilled:

(1)  for any ball lying in $S_T$, all conditions of Sobolev's embedding theorem in a space of continuous functions are certainly valid ([23], p. 64);

(2)  at least, all derivatives of the order up to $m - 1$ satisfy also all conditions Sobolev's theorem from above.

Since an integer number $m$ is an arbitrary, then a weak solution $u$ belongs to the class $C^\infty((0, T_0) \times R^3)$. A smoothness of function $P$ we obtain from Lemma 25 and the smoothness of vector field $u$. The continuity is proved in item 1.

2. Let initial data $\varphi \in C^\infty_{6/5, 3/2}$. We take a test-function $\eta \in C^\infty(R^3)$ such that $\eta(x) = 1$ if $|x| \le 1$ and $\eta(x) = 0$ if $|x| \ge 2$. Consider a solenoidal vector field

$$\Phi^r(x) = \eta(x/r)\varphi(x) - \nabla Q(x).$$

Then, $\triangle Q(x) = \frac{1}{r}\eta_{,i}(x/r)\varphi_i(x)$. A function $Q$ is Poisson's integral

$$Q(x) = -r^{-1} I_2(\eta_{,i}(\cdot/r)\varphi_i)(x).$$

Hence, we have:

$$|\nabla Q(x)| \le r^{-1} I(|\nabla \eta(\cdot/r)| \cdot |\varphi|)(x) \le M r^{-1} I_1(|\varphi|)(x)$$

where $I_1$ is the Riesz potential, $M$ is the maximal mean of $|\nabla \eta|$. From Lemma A1, we obtain

$$\|\nabla Q\|_2 \le A M r^{-1} \|\varphi\|_{6/5} = O(r^{-1}).$$

Direct calculations yield:

$$\|\Phi^r\|_2^2 = \int \left( r^{-2}|\nabla\eta(x/r)|^2 + \eta^2(x/r)|\nabla\varphi(x)|^2 + 2r^{-1}\eta\eta_{,k}(x/r)\varphi_i(x)\varphi_{i,k}(x) \right)dx +$$

$$\|\nabla Q\|_2^2 - 2r^{-1}\int \left( \eta_{,k}(x/r)\varphi_j(x)Q_{,jk}(x) + \eta(x/r)\varphi_{j,k}(x)Q_{,jk}(x) \right)dx.$$

Without the second term in the first integral, the rest of the integrals of all terms in the right-hand tend to zero as $r \to \infty$. This is guaranteed by a test-function $\eta$ and a boundedness of the second derivatives $Q_{,jk}$. The last follows from representation of function $Q$ by Poisson's integral and definition of the class $C^\infty_{6/5,\,3/2}$. In this case, we have two equalities:

$$Q_{,kj}(x) = \frac{1}{4\pi r}\int_{R^3}(\eta_{,i}(y/r)\varphi_i(y))_{,k}\frac{x_j - y_j}{|x-y|^3}dy, \tag{53}$$

$$Q_{,kj}(x) = c_{kj}r^{-1}\eta_{,i}(x/r)\varphi_i(x) + r^{-1}T_{kj}(\eta_{,i}(\cdot/r)\varphi_i)(x), \tag{54}$$

where $c_{kj}$ are universal constants, and $T_{kj}$ are singular integral operators. Therefore, as $r \to \infty$, then

$$\|\nabla\Phi^r\|_2 \to |\nabla\varphi\|_2. \tag{55}$$

A vector field $\Phi^r \in C^\infty_{6/5,\,3/2}$. A summability of the vector field and its derivatives follows from (53) and (54), the equality

$$Q_{,j}(x) = c_{ij}\eta(x/r)\varphi_i(x) + T_{ij}(\eta(\cdot/r)\varphi_i)(x)$$

and Lemma A1. In addition, $\Phi^r \to \varphi$ in the space $J_0^2(R^3)$, $D^\alpha\Phi^r \to D^\alpha\varphi$ in the space $L_2(R^3)$. Laplacians supports $\triangle\Phi^r$ are compact sets. Therefore, there exist solutions $u^r$ and $P^r$ with an initial data $u^r(0,x) = \Phi^r(x)$ satisfying theorem with the number

$$T_0(r) = \left(\frac{9}{4}\right)^4 \frac{\nu^3}{\|\Phi^r\|_2^4}.$$

From (55), we have $T_0(r) \to T_0$ as $r \to \infty$. Fix a number $T < T_0$. From the remark at the beginning of the proof, we conclude all estimates of the theorem for solutions $u^r$, $P^r$. They are uniform with respect to $r$ for $r > r_0$. Hence, sets of mappings $(u^r)_{r>r_0}$, $(P^r)_{r>r_0}$ are bounded in spaces $W_2^1(S_T)$ and $W_6^1(S_T)$. Extract subsequences $(u^{r_k})_{k=1,\dots}$, $(P^{r_k})_{k=1,\dots}$, which converge weakly. Let $u$ and $P$ be its weak limits, respectively. These limits satisfy the next properties:

(1) Lemma 21 is true for them (this is verified in the same way as the proof of Lemma 21 for subsequences);

(2) Lemma 25 is true for them;

(3) Lemma 26 is fulfilled for them. Thus, $u$ and $P$ are weak solutions of problems (1) and (2). Lemma 27 and Lemma 29 are true for vector field $u$. Conditions of growth for a mapping $\varphi \in C^\infty_{6/5,\,3/2}$ show correctness of Lemma 28 for weak solutions from above.

Furthermore, we realize the proof from the first part (see item (1) above). Therefore, the theorem is true also in this case. Theorem 1 is proved.

## 3. Homotopic Property of Cauchy Problem Solutions in Class $C^\infty_{6/5,\,3/2}$

If initial data $\varphi \in C^\infty_{6/5,\,3/2}$, then the Cauchy problem solutions from Theorem 1 have the next homotopic property.

**Theorem 2.** *Let u and P be solutions of problems* (1) *and* (2) *from Theorem* 1. *Then, for every fixed mean* $t \in (0, T_0)$ *(see* (5)*) mappings u, P, $D_t u \in C^\infty_{6/5, 3/2}$. Moreover, all norms*

$$\|u\|_{6/5}, \|\nabla u\|_{3/2}, \|D^\alpha u\|_r, \|D^\beta P\|_r, \|D^\beta D_t u\|_r$$

*if $r > 1$, $|\alpha| \geq 2$, $|\beta| \geq 0$, are uniformly bounded on every segment $[0, T]$ where $T < T_0$.*

Proof of Theorem 2 (it is given below) is relied on for the next simple properties of mappings $v \in C^\infty_{6/5, 3/2}$. For every vector field $v$ and its derivatives of the first order, these are true for both (A1) and representation (Riesz's formula):

$$v(x) = \frac{1}{4\pi} \int_{R^3} \frac{v_{,j}(y)(x_j - y_j)dy}{|x - y|^3}, \quad v_{,j}(x) = \frac{1}{4\pi} \int_{R^3} \frac{v_{,jk}(y)(x_j - y_j)dy}{|x - y|^3}. \tag{56}$$

The second equality we obtain by application of the Stokes theorem to the integral from (56) calculating over a spherical layer $\varepsilon \leq |y - x| \leq r$. From Lemma A4,

$$\int_{|y-x|=r} \frac{|v_{,}(y)|dS}{r^2} \to 0$$

as $r \to \infty$ since $\nabla v \in W^1_{3/2}(R^3)$. Then, the passage to limit as $r \to \infty$, $\varepsilon \to 0$ implies the second equality (56). The first equality is proved in the same way.

We have

$$|v_{,j}(x)| \leq \frac{\pi}{2} I_1(|\nabla v_{,j}|)(x),$$

where $I_1$ is the Riesz potential from (4). Hardy–Littlewood–Sobolev's inequality (see Lemma A1) implies

$$\|I_1(\nabla v_{,j})\|_q \leq A \|\nabla v_{,j}\|_p,$$

where $\frac{1}{q} = \frac{1}{p} - \frac{1}{3}$, $1 < p < q$. Consider only $p \in (1, 3)$. Two last estimates yield $\nabla v \in L_q(R^3)$ for every $q > 3/2$. Analogously with the above, we show for the mapping $v$ and a number $q \in [3/2, 3)$ the belonging $v \in L_r(R^3)$ whenever $r \geq 3$. The logarithmic convex of norm $\|v\|_p$ and Lemma A5 yield norm finiteness $\|v\|_p$ for $p \geq 6/5$. Thus, we proved the next statement.

**Lemma 32.** *Let $v \in C^\infty_{6/5, 3/2}$. Then, $v \in L_p(R^3)$, $\nabla v \in L_q(R^3)$ whenever $p \geq 6/5, q \geq 3/2$.*

**Remark 5.** *Write Poisson's formula (the representation by Riesz's integral $I_2$) for mappings $v$, $\nabla v$ and $D^\alpha v$. Then, we have a boundedness of every vector field $v \in C^\infty_{6/5, 3/2}$ and its derivatives.*

Let

$$P(x) = \frac{1}{4\pi} \int_{R^3} \frac{v_{i,j}(y)v_{j,i}(y)dy}{|x - y|} \tag{57}$$

(the repeated index gives summation).

**Lemma 33.** *Let $v \in C^\infty_{6/5, 3/2}$ and div $v = 0$. Then, the function P and all its derivatives belong to the space $L_r(R^3)$ whenever $r > 1$.*

**Proof.** The integral from (57) we integrate by parts twice over a spherical layer $\varepsilon \leq |y - x| \leq r$. Lemma A4 and the passage to the limit as $r \to \infty$, $\varepsilon \to 0$ imply:

$$P(x) = \frac{|v(x)|^2}{3} - T_{ij}(v_i v_j)(x), \tag{58}$$

where $T_{ij}$ is a singular integral operator with a kernel

$$k_{ij} = \frac{\partial^2}{\partial y_i \partial y_j} \frac{1}{4\pi|x-y|}.$$

Lemma A1 and well-known Calderon–Zygmund's theorem give a summation of function $P$ for any finite exponent $r > 1$. Since

$$P_{,k}(x) = -\frac{1}{4\pi} \int_{R^3} \frac{v_{i,j}(y)v_{j,i}(y)(x_k - y_k)dy}{|x-y|^3},$$

then, analogously with the above, we get:

$$P_{,k}(x) = -\frac{1}{3}v_j(x)v_{k,j}(x) + T_{ik}(v_j v_{i,j})(x). \tag{59}$$

Hence, we obtain a summability of $\nabla P$ whenever finite $p > 1$. A summability of the other derivatives follows from equalities:

$$D^\beta P_{,k}(x) = -\frac{1}{3}D^\beta(v_j v_{k,j})(x) + T_{ik}(D^\beta(v_j v_{i,j}))(x). \tag{60}$$

□

**Lemma 34.** *If $v \in C^\infty_{6/5,\,3/2}$, then Poisson's and Riesz's formulae are true:*

$$v(x) = -\frac{1}{4\pi} \int_{R^3} \frac{\triangle v(y)dy}{|x-y|}, \quad v(x) = \frac{1}{4\pi} \int_{R^3} \frac{v_{,j}(y)(x_j - y_j)dy}{|x-y|^3}.$$

**Lemma 35.** *Suppose a function $P$ and all its derivatives are summaable in space $R^3$ whenever $r > 1$. Let $v, w \in C^\infty_{6/5,\,3/2}$. Then,*

$$\int_{R^3} v_k \triangle w_k dx = -\int_{R^3} v_{k,j}w_{k,j}dx, \quad \int_{R^3} w_k P_{,k}dx = -\int_{R^3} P\,div\ wdx.$$

**Proof.** Apply the second representation from Lemma 34 and make the commutation of integrals. Then,

$$\int_{R^3} v_k \triangle w_k dy = \frac{1}{4\pi} \int_{R^3} v_{k,j}(y) \int_{R^3} \frac{\triangle w_k(y)(x_j - y_j)dx}{|x-y|^3}dy = -\int_{R^3} v_{k,j}w_{k,j}dy$$

(see the first equality of Lemma A4). Changing of integration order is possible because the integral

$$J = \int_{R^6} \frac{|\nabla v(y)||\triangle w(x)|dxdy}{|x-y|^2}$$

is a finite. Really, we have

$$J = \gamma(1) \int_{R^3} |\triangle w| I_1(|\nabla v|)dx.$$

Then, a finiteness follows from a summability of the Riesz potential $I_1(|\nabla v|)$ with exponent 3 (see A1) and the summability of $\triangle w$ with exponent $3/2$. The first equality is proved. To prove the second formula, we observe a finiteness of integrals

$$J_1 = \int_{R^6} \frac{|w(x)||\triangle P(y)|dxdy}{|x-y|^2},$$

$$J_2 = \int_{R^6} \frac{|div\ w(x)||\triangle P(y)|dxdy}{|x-y|} = 4\pi \int_{R^3} |divw| I_2(|\triangle P|)dx.$$

Thereupon, we have:

$$\int_{R^3} w_k P_{,k} dx = \frac{1}{4\pi} \int_{R^3} w_k(x) \int_{R^3} \frac{\triangle P(y)(x_k - y_k)dy}{|x-y|^3} dx =$$

$$\frac{1}{4\pi} \int_{R^3} \triangle P(y) \int_{R^3} \frac{\triangle w_k(x_k - y_k)dx}{|x-y|^3} dy = \frac{1}{4\pi} \int_{R^3} \triangle P(y) \int_{R^3} \frac{div\ w(x)dx}{|x-y|}.$$

$\square$

**Proof of Theorem 2.** Items (1), (2) and (3) from Theorem 1, Lemma 27 and Lemma 28 yield a finiteness of mixed norms $\|D^\alpha u\|_{p_1,\,q_1}$ where $p_1 = \frac{p}{|\alpha|+1}$, $q_1 = \frac{q}{|\alpha|+1}$, whenever $p,\ q \geq 2$, $\frac{3}{p} + \frac{2}{q} \leq 1$. For derivatives of the second order, in particular, we have a finiteness of norm $\|\triangle u\|_{6/5,\,4}$ (see 30). Integrate (1) over segment $[0,t]$ where $t \leq T < T_0$. The solution $P$ is represented by (57). Then, from (59), we get

$$u_k(t,x) - \varphi_k(t,x) = \tag{61}$$

$$\int_0^t (\nu \triangle u_k(\tau,\ x) + T_{ik}(u_j u_{j,\,i})(\tau,x) - \frac{2}{3} u_j(\tau,\ x)u_{j,\,i}(\tau,x))d\tau.$$

Estimate norms in $L_{6/5}$ of every term in (61) in the usual way. We apply Hölder's inequality to interior and exterior integrals. Then,

$$\int_{R^3} \left| \int_0^t u_i u_{k,\,i} d\tau \right|^{6/5} dx \leq t^{1/5} \int_{S_t} u_i u_{k,\,i} d\tau dx \leq \tag{62}$$

$$T^{1/5} \left( \int_{S_t} |u|^2 d\tau dx \right)^{3/5} \left( \int_{S_t} |\nabla u_k|^3 d\tau dx \right)^{2/5},$$

$$\int_{R^3} \left| \int_0^t \triangle u_k d\tau \right|^{6/5} dx \leq t^{1/5} \int_0^t \|\triangle u_k\|_{6/5}^{6/5} d\tau \leq \tag{63}$$

$$T^{9/10} \left( \int_0^T \|\triangle u_k\|_{6/5}^4 d\tau \right)^{3/10} < \infty.$$

The singular integral operator $T_{ik}$ is bounded. Hence, from (61)–(63) and item (2) of Theorem 1, we obtain a uniform estimate of norm $\|u\|_{6/5}$ with respect to $t \in [0,T]$.

In the same way, we prove a summability of gradient $\nabla u$ with any exponent $p \geq 3/2$. From Lemma 27 and Lemma 28, whenever $p,q \geq 2$, $\frac{3}{p} + \frac{2}{q} \leq 1$, we get a finiteness of mixed norms $\|D^\alpha u\|_{p_1,\,q_1}$ for derivatives of the third order where $p_1 = \frac{p}{4}$, $q_1 = \frac{q}{4}$ since $\alpha = 3$, $j = 0$. In particular, we have a finiteness of norm $\|\triangle \nabla u\|_{3/2,\,3/2}$.

Let us differentiate (1) with respect to $x_m$. Thereupon, we integrate its over $[0,t]$ where $t \leq T < T_0$. Formulae (57) and (60) yield

$$u_{k,\,m} - \varphi_{k,\,m} = \int_0^t (\nu \triangle u_{k,\,m} + T_{ik}((u_j u_{j,\,i})_{,\,m}) - \frac{2}{3}(u_j u_{j,\,i})_{,\,m})d\tau. \tag{64}$$

Hence, for exponent $p = 3/2$, we obtain estimates, which are similar estimates (62) and (63). A boundedness $u$, uniform estimates of norm $\|\nabla u\|_3$ (see item (2) from Theorem 1) on segment $[0,T]$, a finiteness of mixed norm $\|\triangle \nabla u\|_{3/2,\,3/2}$ give a uniform boundedness of norms $\|\nabla u\|_{3/2}$.

Let derivative order $|\alpha| \geq 2$. Then, (62) takes the form:

$$D^\alpha u_k - D^\alpha \varphi_k = \int_0^t (\nu \triangle D^\alpha u_k + T_{ik}(D^\alpha(u_i u_{j,\,i})) - \frac{2}{3} D^\alpha(u_j u_{j,\,i}))d\tau. \tag{65}$$

Fix an exponent $r > 1$. Choose numbers $p, q = (|\alpha| + 3)r$. Then, we have a finiteness of the mixed norm $\|D^\alpha u\|_{r, r}$. It follows

$$\left| \int_{R^3} \int_0^t \triangle D^\alpha u d\tau dx \right|^r \leq t^{r-1} \int_{S_t} |\triangle D^\alpha u|^r d\tau \leq T^{r-1} \int_{S_T} |D^\alpha \triangle u|^r d\tau dx < \infty. \tag{66}$$

Terms in derivative $D^\alpha(u_i u_{k, i})$ without coefficients have a form: $D^\beta u_i D^\gamma u_{k, i}$, where $|\beta| + |\gamma| = |\alpha|$. Then,

$$\left| \int_{R^3} \int_0^t D^\beta u_i D^\gamma u_k d\tau dx \right|^r \leq T^{r-1} \int_{S_T} |D^\beta u|^r |D^\gamma \nabla u|^r d\tau dx = T^{r-1} J. \tag{67}$$

To the right-hand side, we apply Hölder's inequality with exponents $\frac{|\alpha|+3}{|\beta|+1}$ and $\frac{|\alpha|+3}{|\gamma|+2}$. Therefore,

$$J \leq \|D^\beta u\|_{p_1, p_2}^{r/p_1} \|D^\gamma \nabla u\|_{p_1, p_2}^{r/p_2},$$

where $p_1 = \frac{p}{|\beta|+1}$, $p_2 = \frac{p}{|\gamma|+2}$. All these mixed norms are bounded. This follows from Lemma 27 and Lemma 28. Hence, formulae (65)–(67) and a boundedness of a singular integral operator give uniform boundedness of all norms with respect to $t \in [0, T]$.

The solution $P$ is represented by (57) with replacing $v$ by $u$. A summability follows from Lemma 33 whenever $r > 1$. Equalities (58) and (59) and a uniform boundedness derivatives norms of vector field $u$ prove a uniform boundedness of norms $\|D^\beta P\|_r$ where $r > 1$. From (1) and proved uniform estimates from above, we have necessary statement for derivative $D_t u$. Theorem 2 is proved. $\square$

## 4. Basic Parameters and Extension of the Cauchy Problem Solutions

Now, we define two from three basic parameters. They have a key part for an extension of the Cauchy problem solutions as solutions with initial data from the class $C^\infty_{6/5, 3/2}$. A functional $l(\varphi)$ and the first parameter $\lambda$ we define by

$$l(\varphi) = \|\varphi\|_2 \cdot \|\nabla \varphi\|_2, \quad \lambda = \left(\frac{4\sqrt[4]{3}}{3a_1}\right)^2 \frac{\nu^2}{l(\varphi)} = \frac{81\nu^2}{8l(\varphi)}, \tag{68}$$

where the constant $a_1$ from Corollary A4. By Theorem 2, the solution of the Cauchy problem with condition $\varphi \in C^\infty_{6/5, 3/2}$ can be extended as the solution in any time $t$. Moreover, extended solutions keep uniform estimates of all norms from Theorem 2 on extended segments $[0, T] \subset [0, T_*)$. In other words, the class $C^\infty_{6/5, 3/2}$ is kept. If $[0, T_*)$ is the maximal interval of solution existence, then the second parameter is defined by:

$$\mu = \frac{T_*}{T_0}, \tag{69}$$

where $T_0$ from (5).

The third parameter $\varepsilon$ is defined below by (87).

*4.1. Solutions Extension in Global with Condition $l(\varphi) < \frac{81\nu^2}{8}$*

**Lemma 36.** *Let $u$ be a solution of problems* (1) *and* (2) *from Theorem 2. Then, functions*

$$\eta_1(t) = \|\nabla u\|_2, \quad \eta_2(t) = \|\triangle u\|_2, \quad \eta_3(t) = \int_{R^3} u_i u_{k, i} \triangle u_k dx, \quad \eta_4(t) = \|u\|_2$$

*are continuous functions on the interval $[0, T_*)$.*

**Proof.** Let $s \in (0, T_*)$. Fix $t \in (s, \tau_1(s)$ where the function $\tau_1$ from Lemma 45. Choose a segment $[T, T_1] \subset (s, \tau_1(s))$ assuming $t$, $t + h \in [T, T_1]$. Denote $z = z(t, h, x) = u(t + h, x) - u(t, x)$. Take equalities (1) with time argument $t + h$. Thereupon, we multiply them by $\triangle z$ getting the scalar product

and integrate over $R^3$. The derivative $D_t u \in C^\infty_{6/5, \, 3/2}$ for all $t \in [0, T_*)$. It follows from Theorem 2.
Then, by Lemma 35 (the scalar product in $L_2$ we write as $(f, g)$), we have:

$$\frac{1}{2} \frac{d}{dh} \|\nabla z\|_2^2 = -\nu(\triangle u(t + h, \cdot), \triangle z) + \int_{R^3} u_i(t + h, x) u_{k, \, i}(t + h, x) \triangle z_k dx.$$

Here, the right-hand side is bounded uniformly (see Theorem 2). Then, $\|\nabla z\|_2^2 = O(h)$ as $h \to 0$.
Triangle inequality implies the continuity of function $\eta_1$.
We write equality (1) for time arguments $t$, $t + h$ and subtract it. Thereupon, the difference we multiply
by $\triangle D_t u(t + h, x)$ getting the scalar product and integrating over the whole space. As a result, we
have

$$(D_t z, \triangle D_t u(t + h, \cdot)) - \nu(\triangle z, \triangle D_h z) +$$

$$\int_{R^3} (z_i u_{k, \, i}(t + h, x) + u_i(t, x) z_{k, \, i}) \triangle D_t u_k(t + h, x) dx = 0.$$

Uniform estimates from Theorem 2 and an integrability for any exponent a.e. imply the equality:

$$\frac{\nu}{2} \frac{d}{dh} \|\triangle z\|_2 = O(1).$$

Then, we have the continuity of function $\eta_2$.
Function continuity of $\eta_3$ follows also from uniform estimates of Theorem 2. Difference $\eta_3(t) - \eta_3(t_0)$
is considered as the sum of three integrals with combinations:

$$u_i(t, \, x) - u_i(t_0, x), \; u_{k, \, i}(t, \, x) - u_{k, \, i}(t_0, x), \; \triangle u_k(t, \, x) - \triangle u_k(t_0, x).$$

Every integral we estimate by Hölder's inequality so that there appear norms:

$$\|u_i(t, \cdot) - u_i(t_0, \cdot)\|_6, \; \|u_{k, \, i}(t, \cdot) - u_{k, \, i}(t_0, \cdot))\|_2,$$

$$\|\triangle u_k(t, \cdot) - \triangle u_k(t_0, \cdot))\|_2.$$

The first of these norms is estimated through the second norm by the inequality from Lemma A1 with
application of the second representation in Lemma 34. Therefore, on every segment $[0, T]$ with some
constants $C_1$, $C_2$, we have:

$$|\eta_3(t) - \eta_3(t_0)| \le C_1 |\eta_1(t) - \eta_1(t_0)| + C_2 |\eta_2(t) - \eta_2(t_0)|.$$

Hence, the first statement follows. Let us prove function continuity of $\eta_4$. The estimate

$$\|u(t, \cdot) - u(t_0, \cdot))\|_6 \le A \|\nabla u(t, \cdot) - \nabla u(t_0, \cdot))\|_2$$

was called above. The logarithmic convex inequality $\|v\|_2 \le \|v\|_{6/5}^{1-\theta} \|v\|_6^\theta$ where $\frac{1}{2} = \frac{5}{6}(1 - \theta) + \frac{\theta}{2}$ and
Theorem 2 (see item (2)) about uniform boundedness of norms) give the statement of the lemma. Here,
it is enough to take $v = u(t, \cdot) - u(t_0, \cdot)$. $\quad \square$

**Lemma 37.** *Let $\varphi \in C^\infty_{6/5, \, 3/2}$ and $l(\varphi) < \left(\frac{v}{a_1}\right)^2$, where $l(\varphi)$ is defined by (68), the number $a_1$ from
Corollary A4. Then, solution $u$ of problems (1) and (2) from Theorem 1 satisfies inequality: $\|\nabla u\|_2 \le \|\nabla \varphi\|_2$.*

**Proof.** Equality (1) we multiply by $\triangle u$ getting the scalar product and integrating over the whole space.
Then, from Theorem 2 and Lemma 35, we have:

$$\frac{1}{2} \frac{d}{dt} \|\nabla u\|_2^2 = \eta_3(t) - \nu \eta_2^2(t),$$

where $\eta_i$, $i = 2, 3$, from Lemma 32. Now, we show that the function $\eta(t) = \eta_3(t) - \nu\eta_2^2(t)$ is negative. Note $\eta(0) < 0$. Suppose the opposite. Then,

$$\nu\|\triangle\varphi\|_2^2 \leq \int \varphi_i\varphi_{k,\,i}\triangle\varphi_k dx.$$

From Corollary A4 (it is extended on the class $C_{6/5,\,3/2}^\infty$ by Lemma 34), we have estimate:

$$\nu\|\triangle\varphi\|_2^2 \leq a_1\|\nabla\varphi\|_2^{3/2}\|\triangle\varphi\|_2^{3/2}.$$

Since

$$\|\nabla\varphi\|_2 \leq \|\varphi\|_2^{1/2}\|\triangle\varphi\|_2^{1/2}$$

(it follows from Lemma 35), then the last two inequalities imply $\nu^2 \leq a_1^2 l(\varphi)$. We have a contradiction. Let $[0, t_0)$ be a maximal interval where function $\eta < 0$. Suppose $t_0 < T_0$. Continuity condition (see Lemma 32 and Theorem 2) gives $\eta(t_0) = 0$.
Repeating arguments from above, we obtain estimate:

$$\nu^2 \leq a_1^2 l(u(t_0, \cdot)). \tag{70}$$

With the other hand, function $\eta_1$ from Lemma 32 is a decreasing function on interval $[0, t_0)$. Therefore, $\|\nabla u(t_0, \cdot)\|_2 < \|\nabla\varphi\|_2$. Since $\|u\|_2 \leq \|\varphi\|_2$, then $l(u(t_0, \cdot)) < l(\varphi)$. Compare this inequality with (70). Then, we have a contradiction. $\square$

**Lemma 38.** *Let* $\varphi \in C_{6/5,\,3/2}^\infty$ *and*

$$q^{\alpha_{m-1}}\left(\frac{\nu}{a_1}\right)^2 \leq l(\varphi) < q^{\alpha_m}\left(\frac{\nu}{a_1}\right)^2,$$

*where* $l(\varphi)$ *is defined by* (68), *numbers* $a_1$ *from Corollary A4,*

$$q = \frac{4}{3}\sqrt[4]{3}, \ \alpha_0 = 0, \ \alpha_m = 2 - \frac{1}{2^{m-1}}, \ m = 1, 2, \ldots.$$

*Then, for solution* $u$ *of problems* (1) *and* (2) *from Theorem 1, there exists a number* $t_0 \in (0, T_0)$ *such that*

$$l(u(t_0, \cdot)) < q^{\alpha_{m-1}}\left(\frac{\nu}{a_1}\right)^2.$$

**Proof.** Suppose the opposite. Then, on interval $[0, T_0)$, the inequality holds:

$$q^{2\alpha_{m-1}}\left(\frac{\nu}{a_1}\right)^4 \leq l^2(u).$$

Integrate it over this interval. Since

$$2\nu\|\nabla u\|_2^2 = -\frac{d}{dt}\|u\|_2^2,$$

then

$$q^{2\alpha_{m-1}}\left(\frac{\nu}{a_1}\right)^4 T_0 \leq -\frac{1}{4\nu}\|u\|_2^4\Big|_0^{T_0}.$$

Take out a nonpositive term on the right-hand side and input the mean $T_0$ from (5). Then,

$$q^{\alpha_m}\left(\frac{\nu}{a_1}\right)^2 \leq l(\varphi).$$

We have a contradiction with the condition.　□

**Theorem 3.** *Let $\varphi \in C^{\infty}_{6/5,\,3/2}$ and $l(\varphi) < \left(\frac{\nu}{a_1}\right)^2$ where $l(\varphi)$ is defined by (68) and the number $a_1$ from Corollary A4. Then, problems (1) and (2) have unique solution $u$ and solution $P$ such that are defined on the set $[0,\infty) \times R^3$. In addition, these solutions have properties (1)–(5) from Theorem 1 on every fixed segment $[0,T]$ and satisfy Theorem 2. Moreover, the norm $\|\nabla u\|_2$, as a function of argument t, is a decreasing function on the set $[0,\infty)$.*

**Proof.** If $T < T_0$, then the statement of theorem follows from Theorems 1, 2 and Lemma 37. A finiteness of mixed norms $\|u\|_{p,\,q}$ we get from a boundedness of the vector field $u$ and estimates $\|u\|_2 \leq \|\varphi\|_2$. Solution uniqueness in the class $L_{p,\,q}$, $\frac{3}{p} + \frac{2}{q} \leq 1$ is proved in [7,8,26] (see also [13]).
Norm monotonicity $\|\nabla u\|_2$ as a function on time argument $t$ follows from condition $\eta < 0$ (see proof of Lemma 37).
Let $[0, T^*)$ be an interval of the maximal length such that there exist solutions with the estimates of Theorem 2.
Suppose $T^* < \infty$. Let $t_0 < T^*$ and $T^* - t_0 < 0,5 T^*$. By Theorem 2 mapping, $u(t_0, \cdot)$ belongs to class $C^{\infty}_{6/5,\,3/2}$. Therefore, by Theorem 1 with this initial data, there is the unique solution $w$ of the Cauchy problem that can be built that can be considered as the extension of solution $u$ (see Lemma 36 and Theorem 1). Extension of $u$ is the unique solution of problems (1) and (2) that satisfies the theorem, at least, on the interval $[0, t_0 + T_2)$, where

$$T_2 = \left(\frac{9}{4}\right)^4 \frac{\nu^3}{\|\nabla u(t_0, \cdot)\|_2^4}.$$

We have $T_2 \geq T_0$ from condition $\|\nabla u\|_2 \leq |\nabla \varphi\|_2$, $w(t,x) = u(t_0 + t, x)$ for means $t < T^* - t_0$. Hence, the solution $u$ is extended with the half-interval $[0, T^*)$ on an interval of more length $[0, T^* + 0,5 T_0)$. We have a contradiction.　□

**Theorem 4.** *Let $\varphi \in C^{\infty}_{6/5,\,3/2}$ and*

$$\left(\frac{\nu}{a_1}\right)^2 \leq l(\varphi) < \left(\frac{4\sqrt[4]{3}\nu}{3a_1}\right)^2 = \frac{81\nu^2}{8},$$

*where $l(\varphi)$ is defined by (68), number $a_1$ from Corollary A4. Then, problems (1) and (2) have a unique solution $u$ and a solution $P$ that are defined on the set $[0,\infty) \times R^3$. In addition, these solutions have properties (1)–(5) from Theorem 1 on every fix segment $[0,T]$ and satisfy Theorem 2. The norm $\|\nabla u\|_2$, as a function of t, is not decreasing function on the set $[T_0, \infty)$, where constant $T_0$ from (5).*

**Proof.** Let $u$ and $P$ be solutions of problems (1) and (2) from Theorem 2. The proof proceeds from induction with respect to number $m$ from Lemma 38. Let $m = 1$. By Lemma 38, there exists a number $t_0 \in (0, T_0)$ such that

$$l(u(t_0, \cdot)) < \left(\frac{\nu}{a_1}\right)^2.$$

By Theorems 1–3, there exists a global solution $w$ of problems (1) and (2) with changed initial data $w(0,x) = u(t_0, x)$. This is the unique smooth extension of solution $u$ that satisfies the proving theorem. Assume the theorem is true for a some natural number $m$. That is, every solution $u$ has a global extension with properties of the theorem if, for this $u$, there exists a number $t_0 \in (0, T_0)$ such that

$$l(u(t_0, \cdot)) < q^{\alpha_m}\left(\frac{\nu}{a_1}\right)^2.$$

Now, we take initial data $\varphi$ such that

$$q^{\alpha_m}\left(\frac{\nu}{a_1}\right)^2 \leq l(\varphi) < q^{\alpha_{m+1}}\left(\frac{\nu}{a_1}\right)^2.$$

By Lemma 38, there exists $t_0 \in (0, T_0)$ satisfying

$$l(u(t_0, \cdot)) < q^{\alpha_m}\left(\frac{\nu}{a_1}\right)^2.$$

By Theorem 2 and the induction hypothesis, there exists a global solution $w$ of problems (1) and (2) with a new initial data $w(0, x) = u(t_0, x)$. By a uniqueness theorem, it is the unique smooth extension of solution $u$ that satisfies the proving theorem. By the induction principle, the theorem is proved because

$$q^{\alpha_m} a_1^{-2} \to \frac{16\sqrt{3}}{9} a_1^{-2}$$

as $m \to \infty$. □

*4.2. Critical $\lambda$ Parameter Mean and the First Hypothetical Turbulent Solution*

Furthermore, it is important in principle an invariant form of a priori estimate for the Cauchy problem solution. An invariance follows from Lemmas 1, 6, 20 and 25, Remark 2, norm semicontinuity of $\|\nabla u\|_2$ and Theorem 1.

**Lemma 39.** *The solution $u$ of problems (1) and (2) from Theorem 1 satisfies estimate:*

$$\|\nabla u\|_2^2 \leq \|\nabla \varphi\|_2^2 (1 - t/T_0)^{-1/2}. \tag{71}$$

**Lemma 40.** *Let $\varphi \in C_{6/5,\,3/2}^\infty$ and $\lambda \geq 1$ i.e.,*

$$l(\varphi) \geq \left(\frac{4\sqrt[4]{3}\nu}{3a_1}\right)^2.$$

*Let $u$ be a solution of problems (1) and (2) from Theorem 1 If*

$$l(u(t_0, \cdot)) < \left(\frac{4\sqrt[4]{3}\nu}{3a_1}\right)^2$$

*for a some number $t_0 \in [0, T_0)$, then solution $u$ can be extended by a global solution with properties (1)–(5) from Theorem 1 and estimates from Theorem 2.*

**Proof.** We construct the extension in the same way as in the proof of Theorem 4. □

**Lemma 41.** *Let $\varphi \in C_{6/5,\,3/2}^\infty$ and parameter $\lambda \leq 1$ (see (68)). If $u$ is the solution of problems (1) and (2) from Theorem 1, then on interval $[0, T_0)$, the inequality holds:*

$$\|\varphi\|_2^2 \left(1 - \lambda^2 + \lambda^2 \sqrt{1 - t/T_0}\right) \leq \|u\|_2^2.$$

**Proof.** We integrate the inequality of Lemma 39 over the segment $[0, t]$. Since

$$\frac{1}{2}\frac{d}{dt}\|u\|_2^2 + \nu\|\nabla u\|_2^2 = 0, \tag{72}$$

then, applying Newton–Leibnitz's formula, we obtain the statement. □

**Lemma 42.** *Let $\varphi \in C^{\infty}_{6/5,\,3/2}$ and parameter $\lambda = 1$ (see (68)). Let u be a solution of problems (1) and (2) from Theorem 1. Suppose, on the interval $[0, T_0)$, there is fulfilled estimate:*

$$l(u(t, \cdot)) \geq \left( \frac{4\sqrt[4]{3}\nu}{3a_1} \right)^2.$$

*Then,*

$$\|u\|_2^2 = \|\varphi\|_2^2 \left( 1 - t/T_0 \right)^{1/2}, \|\nabla u\|_2^2 = \|\nabla \varphi\|_2^2 \left( 1 - t/T_0 \right)^{-1/2}. \tag{73}$$

*If $\lim_{t \uparrow T_0} \|u\|_2 > 0$, then there exists a number $t_0 \in (0, T_0)$ such that*

$$l(u(t_0, \cdot)) < \left( \frac{4\sqrt[4]{3}\nu}{3a_1} \right)^2.$$

**Proof.** Both parts we raise to the second power and integrate over the interval $[0, T_0)$. From (72), we get:

$$\left( \frac{4\sqrt[4]{3}\nu}{3a_1} \right)^4 T_0 \leq \int_0^{T_0} \|u\|_2^2 \|\nabla u\|_2^2 dt = \frac{1}{4\nu} \left( \|\varphi\|_2^4 - \lim_{t \to T_0} \|u\|_2^4 \right), \tag{74}$$

where the number $T_0$ from Theorem 1. Since $\lambda = 1$, then

$$\left( \frac{4\sqrt[4]{3}\nu}{3a_1} \right)^4 T_0 = \frac{\|\varphi\|_2^4}{4\nu}.$$

Therefore, the limit in (74) is equal to zero because, in (74), it must be equalities. This is possible only if, on the interval $[0, T_0)$ (see Lemma 36), it is fulfilled:

$$l^2(u(t, \cdot)) = \left( \frac{4\sqrt[4]{3}\nu}{3a_1} \right)^4 T_0. \tag{75}$$

Integrate (72) over the interval $[0, T_0)$. As the result, we have:

$$2\nu \int_0^{T_0} \|\nabla u\|_2^2 dt = \|\varphi\|_2^2$$

(we take into consideration in formula (74) the limit vanishes ). Apply the estimate of Lemma 39. Then,

$$4\nu T_0 \|\nabla \varphi\|_2^2 \geq \|\varphi\|_2^2.$$

Hence, we have the inequality $\lambda \geq 1$. Since $\lambda = 1$, then the inequality from Lemma (71) must be as the equality. The second formula of lemma is proved. The first follows from (75) and condition $\lambda = 1$. The last statement of lemma we prove from the opposite in the same way. $\square$

**Lemma 43.** *Let initial data $\varphi \in C^{\infty}_{6/5,\,3/2}$, $\varphi \neq 0$, and parameter $\lambda = 1$. There doesn't exist solution u of problems (1) and (2) satisfying (73). It is always true inequality $\lim_{t \uparrow T_0} \|u\|_2 > 0$.*

**Proof.** If such solution exists, then, from (73), we obtain

$$\frac{1}{2} \frac{d}{dt} \|\nabla u\|_2^2 = \left( \frac{8}{81} \right)^2 \frac{\|\nabla u\|_2^6}{\nu^3}.$$

Here, the identical equality is impossible because, for any solution $u$, the inequality (see (7)) is fulfilled:

$$\frac{1}{2} \frac{d}{dt} \|\nabla u\|_2^2 + \nu \|\triangle u\|_2^2 \leq a_1 \|\nabla u\|_2^{3/2} \|\triangle u\|_2^{3/2}.$$

Apply estimates from the proof of Lemma 1. Then, we obtain:

$$\frac{1}{2}\frac{d}{dt}\|\nabla u\|_2^2 \leq \left(\frac{8}{81}\right)^2 \frac{\|\nabla u\|_2^6}{\nu^3}.$$

Compare this inequality with the identity above. Therefore, we must have the equalities for intermediate estimates of Corollary A4 and Lemma 1. Since we used Cauchy–Bunyakovskii's inequality in the Hilbert space $L_2(R^3)$, then there exists a constant $c$ such that

$$u_{i,j} = c\left(u_{k,i}u_{k,j} - \frac{\delta_{ij}}{3}|\nabla u|_2^2\right)$$

for any $i, j = 1, 2, 3$. Hence, we have $u_{i,j} = u_{j,i}$ for each pair $i$, $j$ and $\triangle u \equiv 0$, respectively. From Lemma A6, it follows $u \equiv 0$—a contradiction. The lemma is proved. $\square$

**Lemma 44.** *Let initial data $\varphi \in C_{6/5, 3/2}^\infty$, $\varphi \neq 0$, and parameter $\lambda < 1$. If $u$ is the solution of problems* (1) *and* (2), *then*

$$\lim_{t \to T_0} \|u\|_2^2 > \|\varphi\|_2^2(1 - \lambda^2).$$

**Proof.** Suppose the opposite. Then, we have the equality in Lemma 41. It implies the second equality from (73). Repeating the proof of Lemma 43, we obtain a contradiction. $\square$

**Lemma 45.** *Let $u$ be a solution of problems* (1) *and* (2) *with initial data $\varphi \in C_{6/5, 3/2}^\infty$, $\varphi \neq 0$. Then, a function*

$$\tau_1(t) = t + \left(\frac{9}{4}\right)^4 \frac{\nu^3}{\|\nabla u\|_2^4}$$

*and a function*

$$\lambda(t) = \left(\frac{4\sqrt[4]{3}\nu}{3a_1}\right)^2 \Big/ \|u\|_2\|\nabla u\|_2$$

*with condition $\lambda(0) = \lambda \geq 1$ are not decreasing functions on the interval $[0, T_*)$ where constant $a_1$ from Corollary A4.*

**Proof.** From Theorem 2, we have safety of class $C_{6/5, 3/2}^\infty$ for every $t \in [0, T_*)$ if $u$ satisfies lemma conditions. The both functions are continuous (see Lemma 36 and Theorem 2). Inequality (84) (see below) is true for any mean $\lambda(0) = \lambda$. Rewrite its in another form:

$$\frac{1}{\|\nabla u\|_2^4}\frac{d}{dt}\|\nabla u\|_2^2 \leq \frac{27a_1^4}{128\nu^3}\|\nabla u\|_2^2 \tag{76}$$

and integrate its over the segment $[t, s]$. Simple transformations give:

$$\|u(t, \cdot)\|_2^2(\lambda^2(t) - 1) \leq \|u(s, \cdot)\|_2^2(\lambda^2(s) - 1). \tag{77}$$

Hence, and from lemma condition, it follows $\lambda(s) \geq 1$ for all $s \in [0, T_*)$. Furthermore, we use inequality $\|u(s, \cdot)\|_2 \leq \|u(t, \cdot)\|_2$ and get the monotonicity of the second function. For the monotonicity, the first function follows from inequality (84) because, in this case, $\tau_1' \geq 0$. $\square$

*4.3. Solutions Extension in Global with Condition $l(\varphi) \geq \frac{81\nu^2}{8}$: Necessary Conditions For Hypothetical Turbulence Solutions*

**Lemma 46.** *Let $\varphi \in C^\infty_{6/5, 3/2}$. Suppose that $l(\varphi) \geq \frac{81\nu^2}{8}$ and parameter $\lambda$ from (68). If $\lambda = 1$ or $\lambda < 1$ and the solution $u$ of problems (1) and (2) from Theorem 2 satisfies*

$$\lim_{t \to T_0} \|u\|_2^2 \geq \|\varphi\|_2^2 \sqrt{1 - \lambda^4}.$$

*Then, there exists a number $t_0 \in (0, T_0)$ such that inequality is fulfilled:*

$$l(u(t_0, \cdot)) < \left(\frac{4\sqrt[4]{3}\nu}{3a_1}\right)^2,$$

*where constant $a_1$ from Corollary A4.*

**Proof.** Assume $\lambda = 1$. Then, the statement follows from Lemmas 42 and 43. Let $\lambda < 1$. Suppose the opposite. Then, we have:

$$\left(\frac{4\sqrt[4]{3}\nu}{3a_1}\right)^2 \leq l(u(t, \cdot)).$$

Hence, and from (72), we get

$$\left(\frac{4\sqrt[4]{3}\nu}{3a_1}\right)^4 \leq \|u\|_2^2 \|\nabla u\|_2^2 = -\frac{1}{4\nu}\frac{d}{dt}\|u\|_2^4. \tag{78}$$

Let

$$\alpha = \left(\frac{4\sqrt[4]{3}}{3a_1}\right)^4 \frac{\nu^5}{\|\varphi\|_2^4}. \tag{79}$$

Integrate (78) over segment $[0, t]$. Then, we obtain:

$$\frac{\|u\|_2}{\sqrt[4]{1 - 4\alpha t}} \leq \|\varphi\|_2. \tag{80}$$

Make the passage to the limit in (80) as $t \uparrow T_0$ and compare the new estimate with the inequality from lemma condition. Taking (5), (68) and (80), we conclude $\lim_{t \to T_0} \|u\|_2^2 = \|\varphi\|_2^2 \sqrt{1 - \lambda^4}$. Consider a function $\beta(t) = \|u\|_2^2 - \sqrt{1 - 4\alpha t}\|\varphi\|_2^2$. It vanishes at boundary points of $[0, T_0]$; moreover, $\beta \leq 0$ (see (80)). Let $I \subset (0, T_0)$ be an interval, where the function $\beta$ vanishes at boundary points and $\beta < 0$ on its interior. Then, there exists a point $t_0 \in I$, where $\beta'(t_0) = 0$. Hence, from (72), we get:

$$\nu\|\nabla u(t_0, \cdot)\|_2^2 = \frac{\alpha\|\varphi\|_2^2}{\sqrt{1 - 4\alpha t_0}}.$$

From (80), we have:

$$\nu\|u(t_0, \cdot)\|_2^2 \|\nabla u(t_0, \cdot)\|_2^2 = \frac{\alpha\|\varphi\|_2^2\|u(t_0, \cdot)\|_2^2}{\sqrt{1 - 4\alpha t_0}} \leq \alpha\|\varphi\|_2^4. \tag{81}$$

Compare the left and right sides of this formula and, after we apply (79). Then,

$$l(u(t_0, \cdot)) \leq \left(\frac{4\sqrt[4]{3}\nu}{3a_1}\right)^2.$$

The hypothesis from proof beginning gives the equality:

$$l(u(t_0, \cdot)) = \left(\frac{4\sqrt[4]{3}\nu}{3a_1}\right)^2.$$

Therefore, in (81), the inequality must be by the equality. Hence, we get $\beta(t_0) = 0$. This goes to a contradiction with the choice of the interval $I$. It implies $\beta = 0$. Hence, $\|u\|_2^2 = \|\varphi\|_2^2 \sqrt{1 - 4\alpha t}$. Respectively, from (72), we have

$$\nu\|\nabla u(t, \cdot)\|_2^2 = \frac{\alpha\|\varphi\|_2^2}{\sqrt{1 - 4\alpha t}}.$$

Multiply these equalities. From (78), we obtain:

$$l(u(t, \cdot)) = \left(\frac{4\sqrt[4]{3}\nu}{3a_1}\right)^2.$$

In particular, by Lemma 36, $l(\varphi) \geq \frac{81\nu^2}{8}$. This is impossible with the considering lemma condition. This contradiction proves the lemma. $\square$

Now, we shall study properties of unextended solutions of problems (1) and (2) if such solutions exist. Let $[0, T_*)$ be an interval of the maximal length, where solutions $u$ and $P$ of problems (1) and (2) have properties from Theorem 2. Then, $T_* \geq T_0$ and $T_* = \mu T_0$ (see (69)). Hence, $\mu \geq 1$. Therefore, J. Leray's estimate from [3] can be given in invariant form in the following statement.

**Lemma 47.** *Let $\varphi \in C_{6/5, 3/2}^\infty$ and $l(\varphi) > \frac{81\nu^2}{8}$. Suppose that $[0, T_*)$ is the maximal interval where solutions $u$ and $P$ of problems (1) and (2) have solution properties from Theorem 2. If this interval is a finite, then the following estimate holds:*

$$\|\nabla u\|_2^2 \geq \sqrt{\frac{1}{\mu}}\|\nabla\varphi\|_2^2 \left(1 - t/T_*\right)^{-1/2}. \tag{82}$$

**Proof.** A function $\eta_1(t) = \|\nabla u\|_2$ is unbounded in some left neighborhood of the point $T_*$. Suppose the opposite. Then, for every point $t_0$, there exists solution $v$ of problems (1) and (2) with initial data $v(0, x) = u(t_0, x)$ which satisfy Theorems 1 and 2.

This solution gives the unique extension $u$ on the interval $[t_0, t_0 + l)$, where $l \geq 9^4\nu^3/(4M)^4$ and $M$ is the supremum of $\eta_1$. A point $t_0$ is an arbitrary, therefore, solutions $u$ and $P$ can be extended on the interval $[0, T_* + l)$. In addition, they have solutions' properties from Theorem 2 on this interval. This contradicts the choice of interval with the maximal length.

Now, we prove estimate (82). For solution $u$, we have:

$$\frac{1}{2}\frac{d}{dt}\|\nabla u\|_2^2 + \nu\|\triangle u\|_2^2 = \int u_i u_{k,i}\triangle u_k dx, \tag{83}$$

which follows from (1). It is true for every mean $t \in [0, T_*)$ by Theorem 2 and Lemma 35 because the solution $u \in C_{6/5, 3/2}^\infty$. The integral representation from Lemma 34 permits to apply Corollary A4 and estimate of the right-hand side in the last equality. Repeating the proof of Lemma 1, we obtain

$$\frac{1}{\|\nabla u\|_2^6}\frac{d}{dt}\|\nabla u\|_2^2 \leq \frac{27a_1^4}{128\nu^3}. \tag{84}$$

Integrate (84) over segment $[t, s]$. Then,

$$\frac{1}{\|\nabla u(t, \cdot)\|_2^4} - \frac{1}{\|\nabla u(s, \cdot)\|_2^4} \leq \frac{27a_1^4}{128\nu^3}(s - t). \tag{85}$$

Replace in (84) $s$ on $s_m$, where $s_m \uparrow T_*$ and $\eta_1(s_m) \to \infty$ as $m \to \infty$. The passage to the limit in (85) gives estimate (82). □

**Lemma 48.** *Let $\varphi \in C^\infty_{6/5,\,3/2}$ and $l(\varphi) > \frac{81\nu^2}{8}$. For finite interval $[0, T_*)$ of the maximal length, the parameter $\mu$ is not greater than the number $\lambda^{-4}$, where $\lambda$ is defined by (68).*

**Proof.** In the inequality

$$2\nu \int_0^{T_*} \|\nabla u\|_2^2 dt \le \|\nabla \varphi, \|_2^2$$

we apply Lemma (79). From (5) and (68), we get the statement. □

**Lemma 49.** *Let $\varphi \in C^\infty_{6/5,\,3/2}$, $l(\varphi) > \left(\frac{4\sqrt[4]{3}\nu}{3a_1}\right)^2 = \frac{81\nu^2}{8}$ and $\lambda$ be a parameter from (68). If the interval $[0, T_*)$ has the maximal finite length and solutions $u$, $P$ of problems (1) and (2) have properties from Theorem 2 on this interval, then unextended solutions satisfy conditions:*

$$\|\varphi\|_2^2 \lambda^2 \sqrt{\mu - t/T_0} + \|u(T_*, \cdot)\|_2^2 \le \|u\|_2^2 \le \tag{86}$$

$$\|\varphi\|_2^2 \left(1 - \sqrt{\mu}\lambda^2 + \lambda^2 \sqrt{\mu - t/T_0}\right).$$

**Proof.** Consider a function

$$\omega(t) = \|u\|_2^2 - \|\varphi\|_2^2 \lambda^2 \sqrt{\mu - t/T_0}.$$

From (5) and (68), (72), we have:

$$\omega'(t) = 2\nu\left(-\|\nabla u\|_2^2 + \|\nabla \varphi\|_2^2 \left(\mu - t/T_0\right)^{-1/2}\right), \ \omega'(t) \le 0.$$

Therefore, $\omega(0) \ge \omega(t) \ge \omega(T_* - 0)$. Hence, it follows the first inequality from (86).

Integrate over $[0, t]$ the inequality of Lemma (79). From (72), we obtain:

$$\|\varphi\|_2^2 - \|u\|_2^2 \ge 4T_*\nu \frac{\|\nabla \varphi\|_2^2}{\sqrt{\mu}}\left(1 - \sqrt{1 - t/T_*}\right).$$

Applying (5), (68) and (69), we get:

$$1 - \frac{\|u\|_2^2}{\|\varphi\|_2^2} \ge \sqrt{\mu}\lambda^2\left(1 - \sqrt{1 - t/T_*}\right).$$

Therefore, we have the second inequality in (86). □

**Theorem 5.** *Set $\varphi \in C^\infty_{6/5,\,3/2}$ and $l(\varphi) > \left(\frac{4\sqrt[4]{3}\nu}{3a_1}\right)^2$ with a constant $a_1$ from A4. Let $u$ be a solution of problems (1) and (2) from Theorem 2. If $u$ satisfies condition*

$$\lim_{t \to T_0} \|u(t, \cdot)\|_2^2 \ge \|\varphi\|_2^2 \sqrt{1 - \lambda^4},$$

*then problems (1) and (2) have global solutions $u$ and $P$. Moreover, they have properties (1)–(5) from Theorem 1 on every segment $[0, T]$, $T > 0$ and satisfy conditions of Theorem 2 there. As a function of argument $t$, the product $\|u\|_2\|\nabla u\|_2$ is a decreasing function on the set $[T_0, \infty)$, where constant $T_0$ from (5).*

**Proof.** Let $t_0$ a number from Lemma 46. Without norm monotonicity, the statement of theorem follows from Theorem 2 and Lemma 40. The product $\|u\|_2\|\nabla u\|_2$ is a decreasing function on the set $[t_0, \infty)$. It follows from Theorem 4. Therefore, the theorem is proved. □

Now, we give one result that is connected with a local solutions' extension. If $\lambda < 1$, then we introduce the third parameter $\varepsilon$, which gives a dissipation quantity of a kinetic energy. It is defined by formula:

$$\lim_{t \to T_0} \|u(t, \cdot)\|_2^2 = \|\varphi\|_2^2 (1 - \varepsilon \lambda^2). \tag{87}$$

We observe from Lemmas 43 and 44 that the parameter $\varepsilon$ satisfies strong inequalities: $0 < \varepsilon < 1$. This is very important for the furthest. The usefulness of this parameter is explained by the following result.

**Theorem 6.** *Suppose initial data* $\varphi \in C_{6/5, \, 3/2}^{\infty}$ *and*

$$l(\varphi) > \left( \frac{4\sqrt[4]{3}\nu}{3a_1} \right)^2 = \frac{81\nu^2}{8},$$

*where* $l(\varphi)$ *is defined by* (68). *If solution u of problems* (1) *and* (2) *from Theorem* 2 *satisfies* (87), *then this solution has an extension on the set* $S_{T_3}$ *where*

$$T_3 = \frac{T_0}{4} \left( \varepsilon + \frac{1}{\varepsilon} \right)^2.$$

*This extension has properties* (1)–(5) *from Theorem* 1 *on every segment* $[0, T] \subset [0, T_3)$.

**Proof.** Now, we consider only that solutions which don't have any global and smooth extension. Take $t = T_0$. From theorem condition and the second inequality of Lemma 49, we obtain: $-\varepsilon \leq -\sqrt{\mu} + \sqrt{\mu - 1}$. Hence, we get: $\mu \geq \frac{1}{4} \left( \varepsilon + \frac{1}{\varepsilon} \right)^2$. Then, the statement of the theorem follows from the definition of parameter $\mu$. The theorem is proved. $\square$

**Lemma 50.** *Suppose* $\lambda < 1$. *A finite mean of parameter* $\mu$ *satisfies inequalities:*

$$\frac{1}{4} \left( \varepsilon + \frac{1}{\varepsilon} \right)^2 < \mu \leq \lambda^{-4}.$$

**Proof.** In the first inequality of Lemma 49, we take $t = T_*$. Then, we get the necessary upper estimate. The strong lower estimate doesn't follow from (71) yet. Let

$$\tau(\varepsilon) = \frac{1}{2} \left( \varepsilon + \frac{1}{\varepsilon} \right).$$

Consider a function

$$\varrho(t) = \|u\|_2^2 - \|\varphi\|_2^2 \left( 1 - \tau(\varepsilon)\lambda^2 + \lambda^2 \sqrt{\tau^2(\varepsilon) - t/T_0} \right).$$

We observe $\varrho(0) = \varrho(T_0) = 0$ (see formula (87)). Hence, there exists a number $\xi \in (0, T_0)$ such that $\varrho'(\xi) = 0$. Then,

$$\|\nabla u(\xi, \cdot)\|_2^2 = \frac{\|\nabla \varphi\|_2^2}{\sqrt{\tau^2(\varepsilon) - \frac{\xi}{T_0}}}$$

or

$$T_0 \tau^2(\varepsilon) = \xi + \left( \frac{9}{4} \right)^4 \frac{\nu^3}{\|\nabla u(\xi, \cdot)\|_2^4} = \tau_1(\xi),$$

where function $\tau_1$ from Lemma 45. Since this function does not decrease then for every $t$, $\xi < t < T_0 \tau^2(\varepsilon)$, we have $T_0 \tau^2(\varepsilon) \leq \tau_1(t)$. Therefore,

$$\|\nabla u\|_2^2 \leq \|\nabla \varphi\|_2^2 \left( \tau^2(\varepsilon) - t/T_0 \right)^{-1/2},$$

which holds for every $t$, $\xi < t < T_0\tau^2(\varepsilon)$. Integrating this inequality over interval $[T_0, \tau^2(\varepsilon)T_0)$ from formula (87), we gather:

$$\|u(T_0\tau^2(\varepsilon)), \cdot)\|_2^2 \geq \|\varphi\|_2^2(1 - \tau(\varepsilon)\lambda^2). \tag{88}$$

If $\mu = \tau^2(\varepsilon)$, then, in formula (86), we must have identical equalities (see (88)). It implies the identity

$$\|\nabla u\|_2^2 = \sqrt{\frac{1}{\mu}}\|\nabla\varphi\|_2^2\left(1 - t/T_*\right)^{-1/2}.$$

Take $t = 0$. Hence, we get $\mu = 1$. This contradicts Theorem 6, from which we obtain $\mu \geq \tau^2(\varepsilon) > 1$ because $0 < \varepsilon < 1$. The last proves the strong lower estimate for $\mu$. $\quad\square$

## 5. Main Results, Existence of Global Regular Solutions, and Sufficient Conditions

Now, we prove the basic result which is described by Theorem 7.

**Theorem 7.** *Let $\varphi \in C^\infty_{6/5,\,3/2}$ be initial data, the parameter $\lambda$ from (68) and the number $T_0$ from (5), the vector field u from Theorem 1. If parameter $\lambda \geq 1$ or in opposite case*

$$\lim_{t\uparrow T_0}\|u(t,\cdot)\|_2^2 \geq \|\varphi\|_2^2\sqrt{1 - \lambda^4}, \tag{89}$$

*then the Cauchy problems (1) and (2) have global solutions $u(t,x) = (u_1(t,x), u_2(t,x), u_3(t,x))$ and $P = P(t,x)$ with the following properties:*

*(1) mappings u and P are uniformly continuous and bounded on a set $S_T$ for every number $T$, $T > 0$;*

*(2) for every numbers $T > 0$, $p \geq 6/5$, $q \geq 3/2$, $r > 1$ and multi-indices $|\alpha| \geq 2$, $|\beta| \geq 0$ all norms*

$$\|u\|_p, \ \|\nabla u\|_q, \ \|D^\alpha u\|_p, \ \|D^\beta P\|_2, \ \|D^\beta D_t u\|_r$$

*are uniformly bounded on the segment $[0,T]$, moreover $\|u\|_2 \leq \|\varphi\|_2$;*

*(3) gradients $\nabla u_i$, $i = 1,2,3$, $\nabla P$ are bounded on the set $S_T$ for every $T > 0$;*

*(4) solution u has a finite mixed norm $\|u\|_{p,q}$ on the set $S_T$ for every $T > 0$ and every pair of exponents $p, q \geq 2$;*

*(5) solutions u, P belong to class $C^\infty((0,T) \times R^3) \bigcap C(S_T)$ i.e., these solutions are classical.*

*If parameter $\lambda \geq 1$, then the function $l(t) = \|u\|_2\|\nabla u\|_2$ is a decreasing function on the interval $[0,\infty)$ . If $\lambda < 1$ and condition (89) is fulfilled then the function $l = l(t)$ is a decreasing function on the interval $[T_0,\infty)$.*

**Proof.** Let $\lambda > 1$. Then, the statement follows from Theorem 4.
Let $\lambda = 1$. In this case, the theorem arises from Lemmas 42, Lemma 43 and Theorem 4. The monotonicity of the function $l$ follows from Lemma 45.
Let $\lambda < 1$ and condition (89) is fulfilled. Then, the statement of the theorem arises from Lemmas 45 and 46, Theorem 5. The theorem is proved. $\quad\square$

**Theorem 8.** *Let $\varphi \in C^\infty_{6/5,\,3/2}$ be initial data, the parameter $\lambda < 1$ (see (68)) a vector field u is a weak solution from the Cauchy problems (1) and (2). If on an interval $[0,T)$ an inequality*

$$\|u(t,\cdot)\|_2^2 \geq \|\varphi\|_2^2\left(1 - \lambda^2\sqrt{\frac{t}{T_0}}\right) \tag{90}$$

*is fulfilled, then the weak solutions $u(t,x) = (u_1(t,x), u_2(t,x), u_3(t,x))$ and $P = P(t,x)$ are regular on interval $[0,T)$ and satisfy Theorem 5.*

**Proof.** Let be $[0, T_*) \subseteq [0, T)$ a maximal interval where the weak solution $u$ is regular. Suppose $T_* < T$. Then, from (86) and (90), we have

$$\|u(t, \cdot)\|_2^2 = \|\varphi\|_2^2 \left(1 - \sqrt{\mu}\lambda^2 + \lambda^2 \sqrt{\mu - \frac{t}{T_0}}\right).$$

Since solution $u$ is regular on the interval $[0, T_0)$, then, by differentiation of the previous identity at point $t = 0$, we obtain $\mu = 1$. From Lemma 50, we have a contradiction. Therefore, $T_* = T$. □

Does there exist weak solution $u$ satisfying opposite inequality (90) if $t > T_0$? It is unknown.

## 6. The Cauchy Problem with Less Smoothness of Initial Data

In addition, the invariant class $C^\infty_{6/5,\,3/2}$ Sobolev space $\overset{\circ}{W}{}^3_2(R^3)$ as the closure of infinitely smooth vector fields is another important invariant class, which satisfy existence condition of global solutions.

Different exceptions for solenoidal vector fields from Sobolev classes $\overset{\circ}{W}{}^3_2(R^3)$ and $W^3_2(R^3)$ were shown in [31]. Therefore, we consider the first space from them.

Let $\varphi \in \overset{\circ}{W}{}^3_2(R^3)$ be a solenoidal vector field. Set $(\varphi^m)_{m=1,\dots}$ a sequence of of finite, solenoidal and infinitely smooth vector fields, which converges to the field $\varphi$ in the space $\overset{\circ}{W}{}^3_2(R^3)$. We observe that $\varphi^m \in C^\infty_{6/5,\,3/2}$. Let $(u^m)_{m=1,\dots}$, $(P^m)_{m=1,\dots}$ be sequences of solutions in the Cauchy problem for Navier–Stokes equations with the initial dates $\varphi^m$. Then, all pairs $u^m$, $P^m$ satisfy all uniform estimates of Lemma 21 on any compact set of the interval $[0, T_0^m)$ where $T_0^m = 9^4 \nu^3 / 4^4 \|\nabla \varphi^m\|_2^4$ since upper bounds in these inequalities depend on a set and $\nu$, $\|\nabla \varphi^m\|_2$, $\|\varphi^m\|_2$. Therefore, on every fixed segment $[0, T] \subset [0, T_0]$, we can take these constants as common for all $u^m$, $P^m$ because $\varphi^m \to \varphi$ in the space $\overset{\circ}{W}{}^3_2(R^3)$. Then, without loss of generality, we assume that the sequence $(u^m)_{m=1,\dots}$ converges weakly in the space $W^1_6(S_T)$ to a field $u_0$. In addition, we suppose that $(\triangle u^m)_{m=1,\dots}$, $(\nabla D_t u^m)_{m=1,\dots}$ and $(\nabla P^m)_{m=1,\dots}$ converge weakly in $L^1_2(S_T)$ to $\triangle u^0$, $\nabla D_t u^0$ and $\nabla P^0$. More generally, weak limits $u_0$, $P^0$ satisfy all conclusions of Lemma 21 and they are weak solutions of problems (1) and (2). From the equality (1) for couple $u_0$, $P^0$ and items (2), (4), (5), (8) differentiating (1), we obtain that distributions $\triangle u^0_{,j}$, $j = 1, 2, 3$, belong to the space $L_2(R^3)$ for almost everywhere $t$. Thus, the class $\overset{\circ}{W}{}^3_2(R^3)$ is invariant similar to the class $C^\infty_{6/5,\,3/2}$. For this case, in the same way, we can define the basic parameters $\lambda$, $\mu$, $\varepsilon$. After that, one should note that the statement of Lemma 36 will be true when initial data $\varphi \in \overset{\circ}{W}{}^3_2(R^3)$. Repeating the proof of Theorem 7, we obtain the following result.

**Theorem 9.** *Let $\varphi \in \overset{\circ}{W}{}^3_2(R^3)$ be initial data, the parameter $\lambda$ from (68) and the number $T_0$ from (5). Let be a vector field $u$ is a weak solution of the Cauchy problems (1) and (2). If parameter $\lambda \geq 1$ or in opposite case*

$$\lim_{t \uparrow T_0} \|u(t, \cdot)\|_2^2 \geq \|\varphi\|_2^2 \sqrt{1 - \lambda^4},$$

*then the Cauchy problems (1) and (2) have global solutions $u(t, x) = (u_1(t, x), u_2(t, x), u_3(t, x))$ and $P = P(t, x)$ with the following properties:*

(1)     *mappings $u$ and $P$ are uniformly continuous and bounded on a set $S_T$ for every number $T$, $T > 0$;*
(2)     *for every numbers $T > 0$, $p \geq 2$, $q \geq 2$, all norms*

$$\|u\|_2, \ \|\nabla u\|_2, \ \|\triangle u\|_2, \ \|\nabla D_t u\|_2, \ \|\nabla P\|_2$$

*are uniformly bounded and mixed norms $\|u\|_{p,\,q}$, $\|D_t \triangle u\|_{2,\,2}$ are finite on segment $[0, T]$;*

(3)   solutions $u$, $P$ belong to class $C^\infty((0,T) \times R^3) \bigcap C(S_T)$ i.e., these solutions are classical.
      If parameter $\lambda \geq 1$, then the function $l(t) = \|u\|_2 \|\nabla u\|_2$ is a decreasing function on the interval $[0,\infty)$ .
      If $\lambda < 1$ and

$$\lim_{t \uparrow T_0} \|u(t,\cdot)\|_2^2 \geq \|\varphi\|_2^2 \sqrt{1 - \lambda^4},$$

then the function $l = l(t)$ is a decreasing function on the interval $[T_0, \infty)$.

**Theorem 10.** *Let $\varphi \in W_2^3(R^3)$ be initial data in problems* (1) *and* (2)*. If parameter $\lambda > 1$, then the solution $u$ from Theorem* 9 *satisfies:*

(1)   *a power of norm $\|u\|_2^4$ is a convex function;*
(2)   *there is fulfilled:*

$$\|\nabla u\|_2^2 \leq \|\nabla \varphi\|_2^2 \frac{\lambda^2}{\lambda^2 - 1}.$$

**Proof.** It follows from Lemma 45 because this lemma is true for solution $u$ from Theorem 9.   □

## 7. Integral Identities for Solenoidal Vector Fields: Dimensions Comparison

Some review and results about integral identities for solenoidal vector fields are given by authors in [32,33]. Here, we reduce one from these identities, which shows the essential distinction for the Navier–Stokes equations between space and plane.

Let $u, v, w : R^n \to R^n$ be any triple of solenoidal vector fields from the class $C_0^2(R^n)$. Denote

$$c_{ki}(u) = u_{k,i} - u_{k,i}, \quad k, i = 1, 2, \ldots, n.$$

**Lemma 51.** *(see [32]) For every triple $u$, $v$, $w : R^n \to R^n$ of solenoidal vector fields from the class $C_0^2(R^n)$, the identity holds:*

$$\int (w_{i,j} + w_{j,i}) c_{ki}(v) c_{kj}(u) dx = - \int w_i (c_{ki}(u) \triangle v_k + c_{ki}(v) \triangle u_k) dx.$$

Hence, it follows (one should take $u = v = w$):

$$\int u_{i,j} c_{ki}(u) c_{kj}(u) dx = - \int u_i c_{ki}(u) \triangle u_k dx.$$

**Corollary 1.** *(see [33].)   If dimension $n = 2$, then every solenoidal vector $u \in C_0^2(R^2)$ satisfies the integral identity:*

$$\int u_i u_{k,i} \triangle u_k dx = 0. \tag{91}$$

Obviously, it implies some interesting applications to the 2D Navier–Stokes and Euler equations (see [32]).

(1)   We deduce a priori estimate for a solution $u$, which is not independent of a viscosity:

$$\|\nabla u\|_2 \leq \|\nabla \varphi\|_2 + \int_0^t \|\nabla f\|_2 dt, \tag{92}$$

      where $f$ is an outer force. This improves essentially Ladyzhenskaya's estimate (see [34]).
(2)   In the case $f = 0$, we have formula (83) and, therefore, the norm $\|\nabla u\|_2$ is a decreasing function.
(3)   We give the new proof of the existence of a global weak solution for the Euler equations in plane in the case when an outer force $f = 0$. In addition, the estimate $\|\nabla u\|_2 \leq \|\nabla \varphi\|_2$ is exact and it does not follows from Judovich's results [35]. This explains "the simplicity" of a motion of an ideal fluid on plane.

**Remark 6.** *Let $n = 2$, $f = 0$. Then, the product $\|\nabla u\|_2 \|u\|_2$ is a decreasing function in any case.*

**Remark 7.** *If dimension $n = 3$, then integral from (91) may be not equal to null.*

For a simple example, there is the vector field with the following coordinates:

$$u_i(x) = \lambda_i^2 (l_i, x) e^{-\frac{1}{2}\left(\frac{x_1^2}{\lambda_1^2} + \frac{x_2^2}{\lambda_2^2} + \frac{x_3^2}{\lambda_3^2}\right)}, \ i = 1, 2, 3,$$

where $l_i$ is the $i$–th vector row of the skew-symmetric matrix. Since

$$\int_{R^n} \sum_{i,\,k=1}^{n} u_i u_{k,\,i} \triangle u_k dx = -\int_{R^n} \sum_{i,\,k,\,j=1}^{n} u_{i,\,j} u_{k,\,i} u_{k,\,j} dx,$$

then simple calculations show

$$\int_{R^3} \sum_{i,\,j=1}^{3} u_i u_{j,\,i} \triangle u_j dx = c \sum_{i \neq k} \lambda_i^2 \lambda_k^4 \sum_j l_{ki} l_{ij} l_{kj}$$

with a constant $c \neq 0$. A coefficient $\sum_j l_{ki} l_{ij} l_{kj}$ may be not equal to zero for fixed different means $k$ and $i$ because there is the linear independence of polynomials $\lambda_i^2 \lambda_k^4 - \lambda_k^2 \lambda_i^4$, $i < k$, $i, k = 1, 2, 3$. It gives a distinct from zero of the integral when we choose a suitable skew-symmetric matrix. Respectively, the right side (see (83)) for dimensions $n \geq 3$ can be taken with a large value implying a positive mean of the difference

$$\int_{R^n} \sum_{i,\,j=1}^{n} u_i u_{j,\,i} \triangle u_j dx - \nu \|\triangle u\|_2^2$$

for $t \simeq 0$. It is possible because we can take a factor for initial data $\alpha \varphi$ or diminish viscosity coefficient $\nu$. This implies a growth of the norm $\|\nabla u\|_2$ for space. Obviously, on the plane, this phenomena does not appear.

## 8. Conclusions

Briefly, the main achievements (see Theorems 7–10) have an obvious physical interpretation and, therefore, it may be interesting for applications. Nevertheless, they are connected with monitoring of blow up.

First of all, no phenomena blow up if parameter $\lambda \geq 1$ or kinetic energy satisfies inequality:

$$\lim_{t \uparrow T_0} \|u(t, \cdot)\|_2^2 \geq \|\varphi\|_2^2 \sqrt{1 - \lambda^4}$$

for $\lambda < 1$.

No phenomena blow up on the time interval $[0, T)$ if kinetic energy satisfies inequality:

$$\|u(t, \cdot)\|_2^2 \geq \|\varphi\|_2^2 \left(1 - \lambda^2 \sqrt{\frac{t}{T_0}}\right)$$

with condition $\lambda < 1$.

Finally, we have the importance of the exact lower estimates for kinetic energy of a fluid flow. It is possible that this is one of the new ways where the interesting problem will be studied.

**Funding:** This research received no external funding.

**Acknowledgments:** I would like to say many thanks to all my reviewers for their remarks and advices.

**Conflicts of Interest:** The author declares no conflict of interest.

## Appendix A

*Appendix A.1. About the Riesz Potentials and Integral Representations*

Some technical results are given.

**Lemma A1.** *(Hardy–Littlewood–Sobolev's inequality ([24], p. 141). Let $I_\alpha(f)$ be Riesz's potential defined by (4). Set $0 < \alpha < n$. Then, there exists a constant $A = A(p,q)$ where $\frac{1}{q} = \frac{1}{p} - \frac{\alpha}{n}$, $1 < p < q$, such that the following inequality holds:*

$$\|I_\alpha(f)\|_q \leq A\|f\|_p.$$

In a special case, we give an estimate for operator norm.

**Corollary A1.** *The inequality $A(6,2) \leq \sqrt[3]{\frac{4}{\pi}}$ is true, i.e., $\|u\|_6 \leq \sqrt[3]{\frac{4}{\pi}}\|\nabla u\|_2$.*

**Proof.** It is sufficient to verify this inequality for smooth and finite mappings. From Riesz's formula, we have:

$$|u(x)|^4 = \frac{1}{\pi}\int_{R^3}\frac{|u(y)|^2 u_i(y)u_{i,j}(y)(x_j - y_j)dy}{|x-y|^3}.$$

Multiply it by $|u(x)|^2$. Then, we make a simple estimate and integrate over space. Hence,

$$\|u\|_6^6 \leq \frac{1}{\pi}\int|u(y)|^3|\nabla u(y)|\int\frac{|u(x)|^2}{|x-y|^2}dxdy.$$

The interior integral we estimate applying Leray's inequality

$$\int\frac{|u(x)|^2}{|x-y|^2}dx \leq 4\|\nabla u\|_2^2$$

(see [4], also [7], p. 24), thereupon we use Hölder's inequality. Then,

$$\|u\|_6^6 \leq \frac{4}{\pi}\|u\|_6^3\|\nabla u\|_2^3.$$

It gives the required estimate. $\square$

Let us make more precise well-known integral representations as Poisson's formula and Riesz's formula for smooth functions with compact support.

**Lemma A2.** *Let $w \in C^2(R^3)\bigcap L_p(R^3)$, $p \geq 1$, be a mapping and its Laplacian $\triangle w$ has a compact support. Then, the equalities hold:*

$$w(x) = -\frac{1}{4\pi}\int_{R^3}\frac{\triangle w(y)dy}{|x-y|}, \; w_{,j}(x) = \frac{1}{4\pi}\int_{R^3}\frac{\triangle w(y)(x_j - y_j)dy}{|x-y|^3}, \tag{A1}$$

$$w(x) = \frac{1}{4\pi}\int_{R^3}\frac{w_{,j}(y)(x_j - y_j)dy}{|x-y|^3}, \tag{A2}$$

*(In (A2), repeated indices give summation.)*

**Proof.** To integral

$$\int_{\varepsilon \leq |x-y| \leq r}\frac{\triangle w(y)dy}{|x-y|},$$

we apply twice the Stokes formula removing integrals over spherical layer and derivatives of the mapping $w$. As the result, we have two integrals over sphere $\mid x - y \mid = \varepsilon$ and two integrals over sphere $\mid x - y \mid = r$. They are:

$$\int_{|x-y|=\varepsilon} \frac{w(y)dS}{\varepsilon^2}, \quad \int_{|x-y|=r} \frac{w(y)dS}{r^2}, \tag{A3}$$

$$\int_{|x-y|=\varepsilon} \frac{w_{,j}(y)(x_j - y_j)dS}{\varepsilon^2}, \quad \int_{|x-y|=r} \frac{w_{,j}(y)(x_j - y_j)dS}{r^2}.$$

The third and the fourth integrals we transform again applying the Stokes formula and getting integrals over balls $\mid x - y \mid \leq \varepsilon$, and $\mid x - y \mid \leq r$, respectively. Every integral must contain Laplacian. Since support of $\triangle w$ is a compact set, then these integrals tend to zero as $\varepsilon \to 0$, $r \to \infty$.

The second integral in (A3) we denote by a symbol $I$. Then,

$$I = r^{-3} \int_{|x-y|=r} w(y)(x_j - y_j) \frac{(x_j - y_j)}{r} dS.$$

The Stokes formula application gives the equality:

$$I = r^{-3} \int_{|x-y| \leq r} (3w(y) + w_{,j}(y)(x_j - y_j))dy. \tag{A4}$$

The second term in (A4) we integrate by parts. Therefore,

$$\int_{|x-y| \leq r} (w_{,j}(y)(x_j - y_j))dy = \frac{1}{2} \int_{|x-y| \leq r} (\triangle w(y)(|x-y|^2 - r^2)dy.$$

The integral from the first term in (A4) we estimate applying the Hölder's inequality. Then,

$$|I| \leq 3r^{-3} \|w\|_p (\sigma_3 r^3)^{1-1/p} + \frac{1}{2r} \int_{|x-y| \leq r} |\triangle w(y)| dy,$$

where $\sigma_3$—is the volume of a unit ball. From compactness of Laplacian support and lemma condition, we obtain that integral $I \to 0$ as $r \to \infty$. The first integral in (A3) tends to the mean $4\pi w(x)$ as $\varepsilon \to 0$. formula (A2) we prove by the same way. $\square$

**Corollary A2.** *A mapping $w$ from Lemma A2 satisfies inequalities: $|\nabla w(x)| \leq C_1 x^{-2}$, $|w(x)| \leq C_2 x^{-1}$ with some constants $C_1$ and $C_2$.*

**Proof.** The first inequality follows from the second representation of Lemma A2 and compactness of Laplacian support. The second estimate follows from the third representation of Lemma A2 because the first estimate from the corollary gives:

$$|w(x)| \leq \frac{C_1}{4\pi} \int_{R^3} \frac{dy}{|y|^2 |x-y|^2}.$$

A change of variables $y = |x|z$ proves the second estimate. $\square$

**Corollary A3.** *Let $v,w : R^3 \to R^3$ be mappings which satisfy conditions from Lemma A2. Then,*

$$\int_{R^3} v_k \triangle w_k dy = -\int v_{k,j} w_{k,j} dy.$$

**Proof.** We apply the Stokes formula to the integral from the left side of this equality. From Corollary A2 on a sphere $|y| = r$, we get the following formula: $v_k w_{k,j} = O(r^{-3})$. A passage to the limit as $r \to \infty$ gives the required equality. $\square$

**Lemma A3.** *Let $P : R^3 \to R$ be a function and $P \in L_r(R^3)$ with some exponent $r > 1$ and distributions $P_{,ij} \in L_1((R^3) \cap L_s(R^3)$ for some exponent $s \in (1, 3/2)$. Then, for function $P$, Poisson's formula (the first equality from (A2)) is true.*

**Proof.** For any smooth function $P$, we verify the integral identity the same way as in Lemma A2 with application of Lemma A4 (see below). A density of smooth functions and Lemma A1 prove the statement in a general case because there is continuity of the Riesz potentials in spaces $L_q$. □

**Lemma A4.** *Suppose that a continuous mapping $w : R^n \to R^n$ belongs to the class $W_p^1(R^n)$, $p > 1$. Then, for any point $x$, an exponent $\alpha$, where $\alpha > (n-1)(1 - 1/p)$,*

$$r^{-\alpha} \int_{|x-y|=r} w(y) dS \to 0$$

*as $r \to \infty$.*

**Proof.** Hölder's inequality implies an estimate:

$$|\int_{|x-y|=r} w(y) dS| \leq (\omega_{n-1} r^{n-1})^{1/q} \left( \int_{|x-y|=r} |w(y)|^p dS \right)^{1/p}. \tag{A5}$$

Here, $\omega_{n-1}$ – is the surface measure of an unit sphere, $q = \frac{p}{p-1}$. Let $J = \int_{|x-y|=r} |w(y)|^p dS$. Then,

$$J = \int_{|x-y|=r} |w(y)|^p \frac{y_j - x_j}{r} \frac{y_j - x_j}{r} dS =$$

$$= \frac{n}{r} \int_{|x-y|\leq r} |w(y)|^p dy + p \int_{|x-y|\leq r} |w(y)|^{p-2} w_k(y) w_{k,j}(y) \frac{y_j - x_j}{r} dy.$$

The second integral on the right-hand side is estimated by application of Hölder's inequality. Furthermore, we replace the integration over a ball by the integration over the whole space. Hence,

$$J \leq \frac{n}{r} \|w(y)\|_p^p + p \|w(y)\|_p^{p-1} \|\nabla w\|_p$$

and $J = O(1)$ as $r \to \infty$. Therefore, from (A5), we have the statement. □

*Appendix A.2. Logarithmic Convexity Inequalities and Its Corollaries*

**Lemma A5.** *([36], p. 21). A function $\beta(p) = \|w\|_p$ is a logarithmic convex function. That is, for exponents $r \geq 1$, $s \geq 1$ with condition $\frac{1}{p} = \frac{1-t}{r} + \frac{t}{s}$, where $t \in [0, 1]$, the inequality $\|w\|_p \leq \|w\|_r^{1-t} \|w\|_s^t$ is fulfilled.*

**Corollary A4.** *Let $u, v, w : R^3 \to R^3$ be a triple of mappings satisfying conditions of Lemma A2. Then, the inequality holds:*

$$|\int_{R^3} u_i v_{k,i} \triangle w_k dy| \leq a \|\nabla u\|_2 \|\nabla v\|_2^{1/2} \|\triangle v\|_2^{1/2} \|\triangle w\|_2$$

*with a constant $a = \sqrt{\frac{4}{\pi}}$. In addition, for a solenoidal vector field $u$, there is a more exact estimate:*

$$|\int_{R^3} u_i u_{k,i} \triangle u_k dy| \leq a_1 \|\nabla u\|_2^{3/2} \|\triangle u\|_2^{3/2}, \ k, i = 1, 2, 3,$$

*where $a_1 = \frac{8\sqrt[4]{12}}{27}$.*

**Proof.** We have estimates:

$$|u_i v_{k,\,i} \triangle w_k| \le |u| \cdot |\nabla v_k| \cdot |\triangle w_k| \le |u| \cdot |\nabla v| \cdot |\triangle w|,$$

which follow from the Cauchy–Bunyakovskii's inequality. Apply Hölder's inequality for three factors. Then,

$$|\int_{R^3} u_i v_{k,\,i} \triangle w_k dy| \le \|\nabla u\|_6 \|\nabla v\|_3 \|\triangle w\|_2. \tag{A6}$$

For each coordinate $u_i \in C_0^\infty$, we have $\|u_i\|_6 \le A\|\nabla u_i\|_2$ where $A = \sqrt[3]{\frac{4}{\pi}}$ (see Corollary A1). A density of smooth functions and Lemma A1 give the required estimate in a general case.

Since $\|u\|_6^2 \le \sum_i \|u_i\|_6^2$ (we apply the Minkovskii's inequality with exponent 3), then $\|u\|_6 \le A\|\nabla u\|_2$. Respectively, we have $\|\nabla v\|_6^2 \le \sum_i \|v_{,\,i}\|_6^2$ and $\|v_{,\,i}\|_6^2 \le \|\nabla v_{,\,i}\|_2^2$, $\sum_i \|\nabla v_{,\,i}\|_2^2 = \sum_i \|\triangle v\|_2^2$.

From Lemma A5 with exponents $p = 3$, $r = 2$, $s = 6$ and number $t = 0, 5$, we obtain: $\|\nabla v\|_3 \le \|\nabla v\|_2^{1/2} |\nabla v|_6^{1/2}$. Then, from inequalities above and formula (A6), we prove the estimate with a constant $a = A^{3/2}$.

Now, we verify the other inequality. For solenoidal vector fields, we get (see Corollary A2):

$$\int_{R^3} u_i u_{k,\,i} \triangle u_k dy = -\int_{R^3} u_{i,\,j} u_{k,\,i} u_{k,\,j} dy = -\int_{R^3} u_{i,\,j} \left( u_{k,\,i} u_{k,\,j} - \frac{1}{3}\delta_{ij}|\nabla u|_2^2 \right) dy,$$

where $\delta_{ij}$ is Kronecker's delta. Applying Hölder's inequality to a pair

$$u_{i,\,j},\ u_{k,\,i} u_{k,\,j} - \frac{1}{3}\delta_{ij}|\nabla u|_2^2,$$

we obtain:

$$|\int_{R^3} u_i u_{k,\,i} \triangle u_k dy| \le \sqrt{\frac{2}{3}}\|\nabla u\|_2 \|\nabla u\|_4^2.$$

Since

$$\left( \int_{R^3} \left( \sum_i |u_{,\,i}|^2 \right)^2 dy \right)^{1/2} \le \sum_i \left( \int_{R^3} |u_{,\,i}|^4 dy \right)^{1/2},$$

then, from the inequality

$$\|f\|_4^2 \le \left( \frac{4}{3\sqrt{3}} \right)^{3/2} \|f\|_2^{1/2} \|\nabla f\|_2^{3/2}$$

for vector fields (see [23], Chapter 2 and [27]), we get the second part of the lemma comparing all estimates from above. □

*Appendix A.3. Vanishing of Harmonic and Biharmonic Functions*

**Lemma A6.** *If a harmonic function* $h : R^3 \to R$ *is represented by sum* $h = h_s + h_3 + h_6$ *where functions* $h_p \in L_p(R^3)$, $p = s$, $3$, $6$, $1 < s \le 2$, *then* $h \equiv 0$.

**Proof.** Without loss of generality, we assume that functions $h_p$ are smooth. Otherwise, we take its average defined by a formula

$$h^\tau(x) = \int h(x + \tau y)\omega(y)dy$$

with a kernel $\omega \in C_0^\infty(R^3)$. In the equality,

$$0 = \int_{|y-x|\le r} \triangle h(y)|x - y|^\beta dy = I,$$

$-1 < \beta < -0,5$, we transform the integral applying the Stokes theorem. Let $x = 0$. Then,

$$I = -\beta \int_{|y| \leq r} h_{,j}(y) y_j |y|^{\beta-2} dy + \int_{|y|=r} h_{,j}(y) |y|^{\beta} \frac{y_j}{r} dS = \beta J_1 + J_2.$$

The integral over surface $J_2 = 0$ since $J_2 = r^\beta \int_{|y| \leq r} \triangle h(y) dy = 0$. Hence, $J_1 = 0$. This integral is transformed in the same way as the integral $I$. From the equality,

$$J_1 = \int_{|y| \leq r} h(y) \frac{\partial}{\partial y_j} \left( y_j |y|^{\beta-2} \right) dy - r^{\beta-1} \int_{|y|=r} h(y) dS,$$

by application of the theorem on the mean value of a harmonic function, we conclude the formula:

$$h(0) = \frac{\beta+1}{4\pi r^{\beta+1}} \left( \int_{|y| \leq r} \frac{h_s(y)}{|y|^{2-\beta}} dy + \int_{|y| \leq r} \frac{h_3(y)}{|y|^{2-\beta}} dy + \int_{|y| \leq r} \frac{h_6(y)}{|y|^{2-\beta}} dy \right). \tag{A7}$$

For chosen means $\beta$, each potential $I^{\beta+1}(h_q)(0)$, $q = s, 3, 6$ is finite (see Lemma A1). The passage to the limit in (A7) as $r \to \infty$ yields the equality: $h(0) = 0$. □

**Lemma A7.** *If a biharmonic function $h : R^3 \to R$ has a decomposition $h = h_s + h_3 + h_6$ where functions $h_p \in L_p(R^3)$, $p = s, 3, 6, 1 < s \leq 2$, then $h \equiv 0$.*

**Proof.** Without loss of the generality, we can replace functions $h_p$ by its averages (see above). Then, every average $h_p^\tau \in W_p^1(R^3)$. Now, we fix the averaging parameter $\tau$. Let $x = 0$ and $1 < \beta < 1,5$. Let us show that function $h$ is a harmonic function. It is sufficient to apply the theorem about the mean value of a harmonic function to $\triangle h$ and use the spherical coordinates. Then, for the average, we have:

$$\triangle h^\tau(0) = \frac{\beta+1}{4\pi r^{\beta+1}} \int_{|y| \leq r} \triangle h^\tau(y) |y|^{\beta-2} dy = \frac{\beta+1}{4\pi r^{\beta+1}} J_3. \tag{A8}$$

The integral $J_3$ is transformed by applying three times of the Stokes theorem: twice to the integrals over volume and once to the integral over surface. As a result, we obtain:

$$J_3 = (\beta^2 - 3\beta + 2) \int_{|y| \leq r} h^\tau(y) |y|^{\beta-4} dy - (\beta-2) r^{\beta-3} \int_{|y|=r} h^\tau(y) dS + r^{\beta-2} \int_{|y| \leq r} \triangle h^\tau(y) dy.$$

Furthermore, we apply again the theorem about a mean value for a harmonic function to the third integral. After that, we input the mean of integral $J_3$ in (A8). Then, we conclude:

$$\frac{\triangle h^\tau(0)}{3} = \frac{1 - \beta^2}{4\pi r^{\beta+1}} \int_{|y| \leq r} h^\tau(y) |y|^{\beta-4} dy + \frac{\beta+1}{4\pi r^4} \int_{|y|=r} h^\tau(y) dS.$$

Here, the integral over the volume set tends to a finite mean as $r \to \infty$. The finiteness of this mean is proved in the same way as in Lemma A6. This implies

$$\frac{\triangle h(0)}{3} = \lim_{r \to \infty} \frac{\beta+1}{4\pi r^{\beta+1}} \int_{|y|=r} h(y) dS.$$

An exponent mean $\beta$ belongs to the interval $(1, 3/2)$. Hence, and from Lemma A4, we obtain $\triangle h^\tau(0) = 0$. Taking assumption about a ball center, we obtain that the function $h^\tau$ is a harmonic function. Then, from Lemma A6, $h^\tau \equiv 0$. Let $\tau \to 0$. Then, $h \equiv 0$. □

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
