# Peer review of "The 3D Navier–Stokes Equations: Invariants, Local and Global Solutions"

_axioms, doi:10.3390/axioms8020041_

Round 1

Reviewer 1 Report

The arising questions how to interpret these mathematical results  what is practical utility of these results in a physiscal setting  and what is their practical utility. It would be logical to consider these questions in the final section (Conclusions), but this section is not in the article. This section is absolutely necessary.  

Author Response

If it is possible I can send new version of Introduction

Reviewer 2 Report

See the attached pdf file.

Author Response

If is possible I can send new version of Introduction

Reviewer 3 Report

The novelty must be stated clearly in the Introduction.

Also a Conclusions section would be helpful for the readers of the journal.

Author Response

 I can send new version of Introduction if it is possible

Round 2

Reviewer 1 Report

Reference should be numbered in the order cited.

This modification is required to publish the paper.